# Treatment Effects in Extreme Regimes

## Abstract

Understanding treatment effects in extreme regimes is important for characterizing risks associated with different interventions. This is hindered by the unavailability of counterfactual outcomes and the rarity and difficulty of collecting extreme data in practice. To address this issue, we propose a new framework based on extreme value theory for estimating treatment effects in extreme regimes. We quantify these effects using variations in tail decay rates of potential outcomes in the presence and absence of treatments. We establish algorithms for calculating these quantities and develop related theoretical results. We demonstrate the efficacy of our approach on various standard synthetic and semi-synthetic datasets.

## 1 Introduction

Understanding the effect of an intervention is a primary objective of causal inference. In the Neyman-Rubin framework (Neyman, 1923; Rubin, 1974), the average treatment effect (ATE) and the conditional average treatment effect (CATE) are the quantities of general interest. These metrics are instrumental for informed decision-making (Abadie & Cattaneo, 2018) and have been widely applied to various areas such as econometrics and epidemiology (Rothman & Greenland, 2005).

Despite their importance, these statistics are based on the *expectation* of the difference of the outcome variables influenced by the intervention. However, since they are focused on the expectation, these quantities do not consider the *extreme* behavior of the treatment effect. In many scenarios, it is equally or perhaps more important to understand the *worst case* behavior of interventions rather than only the expected behavior. For example, we may prescribe a policy that is beneficial on average, yet it increases the likelihood of extremely adverse events that we want to completely prevent. This is particularly relevant at an individual level, where an intervention can lead to severe adverse effects for a certain subset of members of the population compared to others. Applied in a medical setting, a specific patient may experience stronger adverse side effects as a response to treatment than those observed on the population level or in the average case. In turn, there is a need to characterize the *maximum* (or similarly the *minimum*) of the treatment effect rather than the average.

This provides the motivation for the present work where we develop a framework for estimating treatment effects in the *tails* of the distribution. Although quantile methods have found application in causal inference (Díaz, 2017; Giessing & Wang, 2021), their performance becomes notably unstable and inaccurate for high quantile statistics, especially when data is limited. Therefore, we use the *extrapolation properties* of Extreme Value Theory (EVT) (Haan & Ferreira, 2006) to propose a measure quantifying extreme treatment effects. Under certain assumptions, EVT provides asymptotic theory for the behavior of data within the tail of a distribution through the Generalized Extreme Value (GEV) distributions. GEVs have location and scale parameters as well as a shape parameter, which is particularly important since it measures the tail decay rate. We propose defining the extreme treatment effect as the difference between the shape parameters of potential outcomes. This intuitively quantifies the change in the distribution tail decay rate upon treatment administration. We refer to these statistics as the *extreme treatment effect* (ETE). Related to this, we also examine the differences in the tails of the individual level potential outcomes referred to as the *conditional extreme treatment effect* (CETE).

**Motivating Example** In applications such as healthcare, it may be necessary to ensure that the tail behavior of an outcome (e.g. corresponding to adverse reactions to a drug) does not significantly change under the treatment intervention. We can use the ETE to characterize which distribution is likely to have more realizations deep in the tail since this quantity describes the difference in the decay rate of the tails. Faster tail decay will correspond to fewer extreme events and vice-versa. The idea is illustrated in Figure 1 where the outcome above 12 corresponds to a serious adverse event but below is not considered harmful. On average, the treatment group appears more beneficial (since it has a negative ATE), but it simultaneously induces a greater risk of serious adverse effects (and thus has a higher ETE). Additionally, we focus on estimating the CETE for the *individual* level rather than the population level, which has implications in fields such as personalized medicine. We illustrate this idea in greater detail in Appendix A. Note that for the problems that we are studying, higher order moments such as variance or skew are insufficient for understanding the tail risk. The reason is due to the fact that many of the underlying phenomena are heavy tailed, and higher order moments may not be defined for these data, resulting in statistics that never converge.

While capturing essential characteristics of the potential outcomes, the proposed ETE and CETE have challenges associated with estimation from data. The first challenge is due to the lack of observed data in the tails of the distributions. By definition, data in the tails are rare, hence we would not expect to have access to many samples from the tail. To counteract this challenge, our framework uses EVT to represent the tail distribution as a parametric class of distributions and extrapolate deep into the tails from near-tail data. The second challenge arises from the unavailability of counterfactual outcomes, which is referred to as the *fundamental problem of causal inference* (Rubin, 1974; Holland, 1986). This challenge persists in any causal inference setting and is due to the fact that we cannot observe the outcome for the same individual under two different treatment regimes. The third challenge is due to *unobserved confounders*. While it is common to assume that the observations are generated such that no unobserved confounders exist, which is known as the unconfoundedness assumption[1], it is often very hard to test this assumption in practice.

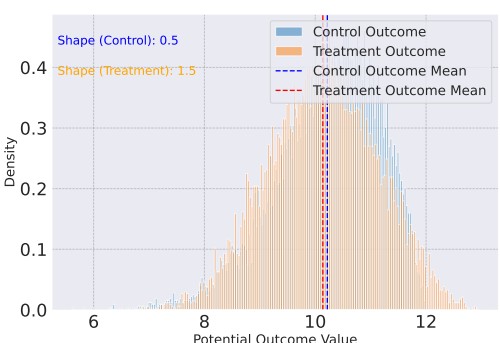

Figure 1: Illustration of what the ETE represents. The control group has faster decaying tails with a shape parameter of 0.5 whereas the treatment group has a shape parameter of 1.5 with slowly decaying tails. Yet, the treatment group has a lower mean. Hence, ETE = 1, while the ATE is negative.

To address these challenges, we propose a new *tail unconfoundedness* condition that we prove to be strictly weaker than unconfoundedness. Moreover, we develop statistical estimators for the proposed ETE and CETE and we prove identifiability and consistency. We illustrate these results numerically in various experiments with both synthetic and semi-synthetic datasets.

**Related Work** As discussed in the introduction, EVT provides a mathematical theory for extrapolating to the tails of a distribution (Haan & Ferreira, 2006). In causal inference, methods for estimating the Average Treatment Effect (ATE) and Conditional Average Treatment Effect (CATE), extreme average treatment effects, and quantile treatment effects have been explored in the literature. Methods for estimating the ATE include covariate adjustment of Rubin (1974), also known as the backdoor adjustment in the do-calculus framework (Pearl, 2009). With regards to deep learning methods for estimating CATE, the *s*-learner and *t*-learner neural networks (Künzel et al., 2019), counterfactual regression (Johansson et al., 2016), and TARNet (Shalit et al., 2017) are well-known examples. Additional machine learning approaches such as random forests (Wager & Athey, 2018) and Gaussian processes (Alaa & Van Der Schaar, 2017) have also been employed for CATE estimation. Other methods have developed doubly robust learners for CATE estimation (Kennedy, 2020). Estimators for extreme quantile treatment effects were developed in (Zhang, 2018), but without considering extrapolation outside the range of observations. Inferring the conditional value at risk of a treatment effect was investigated in Kallus (2023) but the case of tail quantiles was not

---

[1] Also referred to as conditional exchangeability.

explicitly considered. Estimating the extreme average treatment effect was addressed in (Huang et al., 2022) using extreme value theory. Works on estimating a parametric or semi-parametric form of the counterfactual distribution include Kennedy et al. (2021); Chernozhukov & Hansen (2005); Chernozhukov et al. (2013). Methods for estimating the quantiles include (Díaz, 2017; Firpo, 2007; Frölich & Melly, 2013; Zhang et al., 2012; Giessing & Wang, 2021). These methods do not consider estimating the conditional extreme treatment effect and focus on the population level. Moreover, we provide a different framework to define these quantities of interest and provide a detailed mathematical motivation that exploits the extrapolating behavior of EVT. Our approach is closely related to the study of max-stable processes, which have been applied across various disciplines to model extremes over function spaces (Davison et al., 2012). The key difference is these distributions are often estimated assuming an abundance of temporal observations, which is an assumption that may not hold in causal inference settings.

## 2 Background

### 2.1 Causal inference

Consider covariates $X \in \mathcal{X}$ with $\mathcal{X} \subset \mathbb{R}^d$ being the domain. Let $T \in \mathcal{T} = \{0, 1\}$ denote the treatment variable. The group assigned $T = 0$ and $T = 1$ are called the control and treatment groups, respectively. Denote the variable $Y \in \mathbb{R}$ as the factual outcome. Potential outcomes are the outcomes that would have been observed if only treatment $T = 1$ or $T = 0$ was assigned and are respectively represented by $Y_1$ and $Y_0$ (Morgan & Winship, 2015). The relationship between the factual outcome and the potential outcomes is $Y = Y_1 T + Y_0(1 - T)$. The main challenge in causal inference is that we do not observe the counterfactual outcomes, i.e., the outcomes that would have been observed if the treatment were reversed. We call any statistics of the potential outcomes a *causal statistics*. A causal statistic is said to be identifiable if it can be reduced to a function of the factual outcomes $Y$, the treatment assignment $T$, and the covariates $X$ (Imbens, 2004). This is formalized in the following definition:

**Definition 2.1** (Causal Identifiability). *A causal statistic (e.g $\mathbb{E}[Y_1]$) is identifiable if it can be computed from a purely observational quantity (e.g. $\mathbb{E}[\mathbb{E}[Y \mid T = 1, X]]$).*

Similarly, an estimator, as a particular statistic, is said to be identifiable when it meets this criterion. We next describe some assumptions that are often made in order to guarantee the causal identifiability of CATE and ATE. The first is consistency, which requests that the potential outcomes align with the factual outcome:

**Assumption 2.2** (Consistency). *For $t \in \mathcal{T}$, the following holds, $T = t \implies Y_t = Y$.*

The next is overlap[2] which requires that the different treatment groups have the same support:

**Assumption 2.3** (Overlap).
$$\forall x \in \mathcal{X}, \ 0 < P(T = 1 \mid X = x) < 1$$

Finally we assume unconfoundedness, which requires that the potential outcomes are independent of the treatment given the covariates:

**Assumption 2.4** (Conditional Unconfoundedness). $(Y_1, Y_0) \perp\!\!\!\perp T \mid X$

We will later relax this assumption to a weaker *tail unconfoundedness* assumption, which intuitively states that this property only needs to hold in the tail of the outcome variable.

### 2.2 Extreme value theory

EVT provides a principled mathematical framework for extrapolating to the tails of a distribution. The Fisher-Tippett-Gnedenko theorem states that if the shifted and scaled versions of maximum order statistics of a distribution $P(\cdot)$ converge to a non-degenerate distribution, then it is guaranteed to converge to a family of well-specified distributions known as the generalized extreme value (GEV) distributions (Haan & Ferreira, 2006; Fisher & Tippett, 1928; Gnedenko, 1943). If this convergence holds, the distribution $P(\cdot)$ is said to

---

[2]Also known as the positivity assumption.

be in the *maximum domain of attraction (MDA)*[3] of the GEV. The distribution is characterized by three parameters: $\mu$, the location parameter; $\sigma$, the scale parameter; and, $\xi$ the *shape* or *tail index* parameter.

**Definition 2.5.** *The GEV distribution is defined as:*

$$G_\xi(y) = \exp\left(-(1 + \xi\bar{y})^{-1/\xi}\right), \ \ 1 + \xi\bar{y} > 0 \tag{1}$$

*where $\bar{y} = \frac{y-\mu}{\sigma}$ with parameters $\mu, \sigma, \xi$ The case $\xi = 0$ is defined as the limit where $\xi$ approaches zero.*

For our purposes the shape parameter $\xi$ is of interest, which governs the rate of tail decay. When $\xi > 0$ the distribution exhibits a heavy tail, whereas when $\xi < 0$ the tail is bounded. If $\xi = 0$, the tail decays exponentially. The Fréchet, Weibull, and Gumbel distributions respectively correspond to positive, negative, and zero values of $\xi$.

**Theorem 2.6** (Fisher–Tippett–Gnedenko)**.** *Let $Y^{(1)}, Y^{(2)}, \ldots, Y^{(n)}$ be a sequence of iid real random variables. If there exist two sequences of real numbers $a_n > 0$ and $b_n \in \mathbb{R}$ such that the following limits converge to a non-degenerate distribution function:*

$$\lim_{n\to\infty} P\left(\frac{\max\left\{Y^{(1)}, \ldots, Y^{(n)}\right\} - b_n}{a_n} \leq y\right) = G(y),$$

*then the limiting distribution $G$ is the GEV distribution.*

In general, fitting such distributions requires many repeated observations such that the maximum can be computed, thus obtaining samples approaching the tail. The *block maxima* method refers to the approach when the data is subdivided into blocks and the maximum quantity per block is used for fitting (Coles et al., 2001). Specifically, $K \times n$ observations $\left\{Y^{(i,j)}\right\}_{i,j=1}^{K \times n}$ observations are divided into $K$ blocks of size $n$. Let $M^{(i)} = \max_{j=1,\ldots,K} Y^{(i,j)}$ for $i = 1, \ldots, n$. The values $\left\{M^{(i)}\right\}_{i=1}^{n}$ are then used to fit the GEV model. In the causal inference setting, multiple observations per individual is often prohibitively expensive to obtain, so we introduce alternatives to the block maxima method later in the text.

## 3 Theoretical Framework

We now present the proposed framework for estimating treatment effects in extreme regimes. We begin by defining the estimation problem motivated by the previous examples. Then, we establish an identifiability theorem under the condition of *tail unconfoundedness*, which is a weaker requirement than conditional unconfoundedness. Combining these results leads to a numerical algorithm that can be readily implemented. The proofs of our results are presented in Appendix B.

### 3.1 Problem setup

Given a dataset $S = \{(x^{(i)}, t^{(i)}, z^{(i)})\}_{i=1}^{N}$, sampled from a $(X, T, Z)$ where $Z$ is the limiting tail distribution for $Y$. Our goal is to estimate the effect of the treatment assignment on the tail indices of the potential outcomes $Y_0$ and $Y_1$ as well as the conditioned potential outcomes $Y_0|X = x$ and $Y_1|X = x$. This estimation faces the major challenge in causal inference that the potential outcomes are only partially observed and are subject to *selection bias*. To address this, we investigate the identifiability of the conditional maximum likelihood estimator under tail unconfoundedness. When presenting the method, we assume that the domain of attraction of each individual can be estimated, and we assume repeated observations for each individual. In practice, only one factual outcome per individual is usually observed, making it difficult to estimate the maxima through standard preprocessing methods such as the block maxima method. We provide a practical way of dealing with this in Appendix C. Let $x \in \mathcal{X}$ and consider the potential outcomes $Y_1$ and $Y_0$. The outcomes conditioned on the individual $X = x$ are denoted as $Y_1(x) = (Y_1 \mid X = x)$ and $Y_0(x) = (Y_0 \mid X = x)$. We assume both the potential outcomes and their conditionals converge to non-degenerate distributions within

---

[3]The MDA of a distribution $p$ is the distribution to which the maximum of samples from $p$ converges to.

the domain of attraction of the Generalized Extreme Value distributions: $\text{GEV}(\underline{\mu}_1, \underline{\sigma}_1, \underline{\xi}_1)$, $\text{GEV}(\underline{\mu}_0, \underline{\sigma}_0, \underline{\xi}_0)$, $\text{GEV}(\mu_1(x), \sigma_1(x), \xi_1(x))$, and $\text{GEV}(\mu_0(x), \sigma_0(x), \xi_0(x))$. Thus, the conditions of the Fisher-Tippet-Gnedenko theorem are satisfied.

**Definition 3.1** (ETE and CETE)**.** *The extreme treatment effect is defined as:*

$$\underline{\tau}_{ext} = \underline{\xi}_1 - \underline{\xi}_0.$$

*The conditional extreme treatment effect is defined as:*

$$\tau_{ext}(x) = \xi_1(x) - \xi_0(x).$$

We introduce two metrics: $\varepsilon_{\text{ETE}}$ which quantifies the error in estimating ETE, and $\varepsilon_{\text{CETE}}$, measuring the precision in estimating CETE.

**Definition 3.2.** *Given estimators $\hat{\underline{\tau}}_{ext}$ for ETE and $\hat{\tau}_{ext}(x)$ for CETE, the performance errors are respectively defined as:*

$$\varepsilon_{ETE} = |\hat{\underline{\tau}}_{ext} - \underline{\tau}_{ext}|$$

*and*

$$\varepsilon_{CETE} = \mathbb{E}\left[ (\hat{\tau}_{ext}(x) - \tau_{ext}(x))^2 \right].$$

For our proposed methods, we will use this precision measure for quantifying the performance of different methods in our empirical studies.

## 3.2 Mathematical Motivation

In this section, we provide two mathematical interpretations of the ETE and CETE to motivate why these quantities are interesting in a causal inference setting.

**Regularly Varying Perspective.** In this perspective, we can relate the ETE and CETE to the different rates of tail decays of a function through regular variation. The basic interpretation is that when both tail indices are positive, the difference between them relates to the ratio of the survival distributions and can be used to study which group is more likely to have outcomes exceed a certain threshold. We review some definitions on regularly varying functions from Resnick (2007, Chapter 1) and refer the reader to the chapter for additional information. A function $U : \mathbb{R}_+ \to \mathbb{R}_+$ is regularly varying $(RV)$ if $\exists \xi \in \mathbb{R}$ such that

$$\forall x > 0, \lim_{t \to \infty} \frac{U(tx)}{U(t)} = x^\xi.$$

This definition is due to Karamata (Bingham et al., 1989). The parameter $\xi$ is called the exponent of variation, and we write $U \in RV(\xi)$ to mean that $U$ is a regularly varying function with exponent of variation $\xi$. We will present the results for the Fréchet case where $\xi > 0$. The following remark characterizes our motivation Resnick (1988)

**Remark 3.3.** *Let $\underline{\xi}_0, \underline{\xi}_1 > 0$. Let $S_1(Y_1), S_0(Y_0)$ be the survival distributions of the potential outcome variables with positive support for the treatment and control groups respectively. Then if $\underline{\tau}_{ext} > 0$, then $S_0(Y_0)/S_1(Y_1) \in RV(\underline{\xi}_1 - \underline{\xi}_0)$. Moreover, $S_1(Y_1) < S_0(Y_0)$ for $Y_t \gg 1$.*

The proof for this remark relies on the Karamata representation theorem (Galambos & Seneta, 1973), with further details in Appendix B. Therefore, the ETE and CETE can be interpreted as characterizing the ratio of the survival functions between the treatment and control groups.

**Point Process Perspective.** Another practical viewpoint of EVT involves interpreting the GEV in terms of the intensity of a Poisson point process (Coles et al., 2001, Chapter 7). The intensity of the Poisson process is given by the negative log of the density in equation 1. One can use this form to calculate the expected number of extreme points within a region of the state space. We formalize this in the following remark:

**Remark 3.4** (Expected Number of Events). *Consider a causal inference task with potential outcomes variable $Y_t$, $t \in \{0,1\}$, that $Y_t$ is in the maximum domain of attraction of a GEV with rate $\underline{\xi}_t$, and ETE given by $\tau_{ext} > 0$. Suppose that $\mathcal{R} \subset \mathbb{R}$ is the Borel set of interest that should be quantified for the causal inference task with $|\inf \mathcal{R}| \gg 0$. Then, $\mathbb{E}[N_{Y_1}(\mathcal{R})] > \mathbb{E}[N_{Y_0}(\mathcal{R})]$ where $N_{Y_t}(\mathcal{R}) = \sum_{i=1}^{\infty} \delta_{Y_t^{(i)} \in \mathcal{R}}$ for $t \in \{0,1\}$ is a counting process over outcomes on some Borel subset of $\mathbb{R}$.*

The analog of this remark can easily be made for the CETE case by conditioning on the individual covariates. This is particularly important, for example, in a medical setting where one may want to consider the number of severe adverse events in a treatment group versus a control group. If the outcome variable being above a certain threshold corresponds to a serious adverse event, then the ETE and CETE can be used to quantify the number of events.

### 3.3 Tail unconfoundedness and identifiablity

In order for the problem of estimating ETE and CETE to be tractable, we need to establish identifiability of the statistics involving counterfactual quantities in Definition 2.1 before these quantities can be estimated from observational quantities. To this end, in addition to Assumptions 2.2 and 2.3, we will introduce a new assumption in order to guarantee identifiability.

**Assumption 3.5** (Tail Unconfoundedness).

$$\forall t \in \mathcal{T}, \ P(Z_t \mid X, T) = P(Z_t \mid X)$$

*where $Z_t$ is the MDA of the potential outcome $Y_t$.*

Note that Assumption 3.5 is a weaker condition than the unconfoundedness presented in Assumption 2.4, which is assumed in ATE and CATE estimation (Imbens, 2004). Assumption 2.4 requires that all the confounding variables are observed. On the other hand, Assumption 3.5 only requires observing the confounding variables that affect the tail. This is an important relaxation since, even if some covariates that do not affect the tail are missing, ETE and CETE can still be estimated.

**Example where tail unconfoundedness is strictly weaker than unconfoundedness.** To illustrate how the relaxation in the assumption is important, we present discrete and continuous examples of distributions that satisfy Assumption 3.5 but not Assumption 2.4. For the discrete example let $P(Y_1 \mid T = t, X)$ be a Bernoulli distribution with a non-binary treatment parameter $t \in (0,1)\}$. For the continuous example let $P(Y_1 \mid X = x, T = t)$ follow a Beta distribution $\mathcal{B}(\alpha, \beta)$ with parameters $\alpha = x$ and $\beta = t + 1$ for $x, t > 0$. Additional details are provided in Appendix D. Moreover, Assumption 2.4 implies Assumption 3.5 formalized in the proposition below.

**Proposition 3.6.** *If unconfoundedness (Assumption 2.4) holds then so does the tail unconfoundedness (Assumption 3.5).*

Next, we define our conditional maximum likelihood estimator for estimating the individual tail indices. We assume that the location and the scale parameters are known. Let $\mathcal{H} : \mathcal{X} \times \mathcal{T} \to \mathbb{R}$ be a class of hypothesis functions estimating the shape parameter $\xi_t(x)$ for $x, t \in \mathcal{X} \times \mathcal{T}$. In this section, we will assume that the location and scale parameters $\mu, \sigma$ are known. In our empirical results, $\mathcal{H}$ is considered to be a class of neural networks with parameters $\theta \in \Theta$, i.e $\mathcal{H} = \{g_\theta : \mathcal{X} \times \mathcal{T} \to \mathbb{R} | \theta \in \Theta\}$.

**Definition 3.7** (CETE Estimator). *For $t \in \{0,1\}$, let $Z_t(x)$ be the conditional random variable $Z_t | X = x$ for the maximum domain of attraction of $Y_t | X = x$. The shape parameter $\xi_t(x)$ estimator is defined as:*

$$\theta^* = \arg\max_{\theta \in \Theta} \sum_{j=1}^{N} \log p\left(Z_t(x^{(j)}) = z_t^{(j)} \mid g_\theta\left(x^{(j)}, t\right)\right)$$

*and for every $x \in \mathcal{X}$,*

$$\hat{\xi}_t(x) = g_{\theta^*}(x, t)$$

*Accordingly, we define our CETE estimator as: $\hat{\tau}_{ext}(x) = \hat{\xi}_1(x) - \hat{\xi}_0(x)$.*

Since $Z_1|X = x$ is given as the tail distribution, $p$ is taken to be the density of the conditional GEV in Definition 2.5. A naive estimator for the ETE can be defined as follows:

$$\tilde{\underline{\tau}} = \tilde{\underline{\xi}}_1 - \tilde{\underline{\xi}}_0.$$

where,

$$\tilde{\underline{\xi}}_t = \arg\max_\phi \sum_{j=1}^N \log p\left(Z = z^{(j)}|T = t; \phi\right)$$

Therefore, the naive estimator takes the samples from the GEV of the outcomes conditioned on the treatment and fits a GEV to it. This results in a biased estimator of the ETE because of the selection bias introduced by the treatment selection. Now we define our proposed ETE estimator based on the the CETE estimator.

**Definition 3.8** (ETE Estimator). *For $t \in \{0, 1\}$, let $\{\hat{\xi}_t(x^{(j)})\}$ be the estimated conditional shape parameter using the CETE estimator. We sample $\{\hat{z}_t^{(j)}\}_{j=1}^N$ from the GEV with the estimated shape parameters. The tail index $\underline{\xi}_t$ estimator is defined as:*

$$\widehat{\underline{\xi}}_t = \arg\max_\phi \sum_{j=1}^N \log p\left(\hat{Z}_t = \hat{z}_t^{(j)}; \phi\right)$$

*Accordingly, we define our ETE estimator as: $\hat{\underline{\tau}} = \hat{\underline{\xi}}_1 - \hat{\underline{\xi}}_0$.*

The following theorem makes the defined causal quantities statistically tractable.

**Proposition 3.9** (Identifiability). *Under Assumptions 2.2, 2.3, and 3.5, the CETE and ETE estimators defined in Definition 3.7 and Definition 3.8 respectively are causally identifiable.*

The identifiability theorem reduces our original causal statistics (that depend on tails of the potential outcomes) to quantities that only depend on observable variables. However, even under an identifiability assumption, the estimator in Definition 3.7 may not be easy to compute due to the fact that only one observation per individual may exist. We include a theoretical discussion as well as empirical results when only one outcome is observed per individual in Appendix C.

We now prove the following statistical inference results for the proposed estimators.

**Remark 3.10.** *The ETE estimator defined in Definitions 3.8 is consistent and asymptotically normal.*

This remark asserts that, with an increasing number of observed data points, the ETE and CETE estimators converge to their true values and their distributions approach a Gaussian.

## 4 Computational Method

In this section, we present the computational method employed in our study. Given a dataset $\{(x_i, t_i, y_i)\}_{i=1}^N$, we first apply a variation of the block maxima method. We then learn the different parameters of the limiting GEV by maximizing the conditional log-likelihood of the maximum of the outcomes to the covariates. The CETE and ETE can then be computed using the estimated quantities on $\xi$, a pseudocode of the algorithm is provided in Appendix E.

**Empirical $\varepsilon$-max-sampler.** As noted, the standard procedure of using the block maxima in time cannot be used for practical scenarios of interest in the causal setting since we do not have a temporal component — we only observe one outcome for every individual. To approach the task of approximating the maximum of an outcome variable across multiple copies, we instead consider the analogy of the block maxima in *space* to exploit the property that individuals with similar characteristics tend to have similar outcomes. As a result, we can use realizations of individuals locally to approximate independent and identical realizations of a single individual. As the number of points increases and the ball size around every individual converges to zero the approximation of multiple copies of individual outcomes becomes increasingly accurate when taking the maximum of their outcomes.

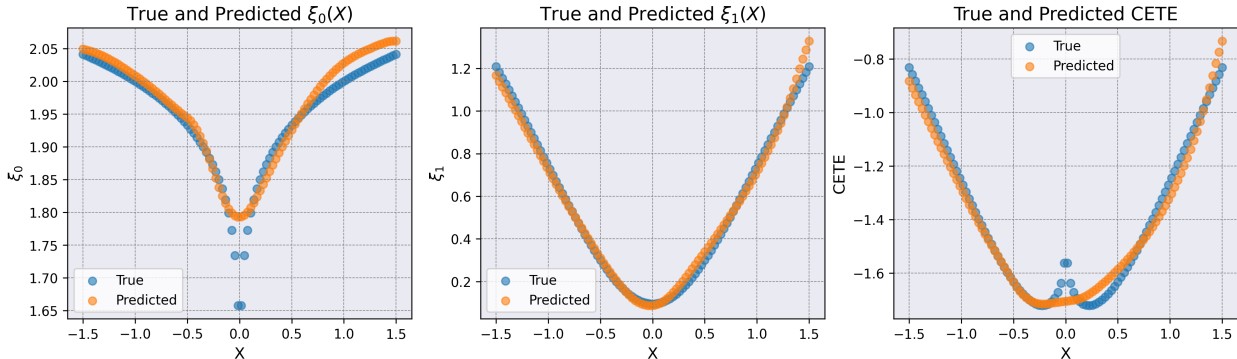

Figure 2: From left to right, the figures represent the estimation of the individual tail parameter under the control condition, the tail parameter under the treatment condition, and the Conditional Extreme Treatment Effect (CETE) using our proposed method.

To do this, we use the $k$-means algorithm to identify clusters of individuals with similar feature profiles. Then, within each cluster, we select the individual with the largest outcome as an approximation of the maximum outcome for that group of similar individuals. We provide a detailed description of the $\varepsilon$-max-sampler procedure in Appendix E as well as a theoretical analysis of this empirical approach in Appendix C.

**Likelihood-Based Estimator.** Since we assume that the potential outcomes are in the maximum domain of attraction of a GEV, we maximize the likelihood of a GEV parameterized by functions mapping the observations to the parameters. Let $\theta_t = (\mu_t, \sigma_t, \xi_t)$ where $\mu_t : \mathcal{X} \to \mathbb{R}$, $\sigma_t : \mathcal{X} \to \mathbb{R}_+$ and $\xi_t : \mathcal{X} \to \mathbb{R}$ for $t \in \{0, 1\}$. Let $\{(x^{(m_i)}, t^{(m_i)}, y^{(m_i)})\}_{i=1}^{K}$ $K$ be the data points after computing the $\varepsilon$-max-sampler. Then we have for $t \in \{0, 1\}$,

$$\hat{\theta}_t = \arg\max_{\theta} \sum_{i=1}^{K} \log(\ell(\theta \mid x^{(m_i)}, t^{(m_i)}, y^{(m_i)})) \mathbb{1}_{(t^{m_i} = t)}$$

where

$$\ell(\theta \mid x, t, y) = \frac{\Phi(y \mid x, t)^{\xi_t(x)+1}}{\sigma_t(x)} e^{-\Phi(y \mid x, t)}$$

and

$$\Phi(y \mid x, t) = \left(1 + \xi_t(x) \left(\frac{y - \mu_t(x)}{\sigma_t(x)}\right)\right)^{-1/\xi_t(x)}.$$

After fitting the parameters, we can estimate the CETE as the following $\hat{\tau}_{ext}(x) = \hat{\xi}_1(x) - \hat{\xi}_0(x)$. In our synthetic experiments, we note that fitting all three conditional parameters of the Generalized Extreme Value (GEV) distribution can be numerically unstable. Given that our primary objective is to investigate the variability of the shape parameter, we assume that the location and scale parameters are known. Our focus is to estimate the shape parameter as a function of the covariates, in order to accurately estimate the CETE and the ETE. It is crucial to highlight that the shape parameter is our parameter of interest since it characterizes the heaviness of the tail—a feature that is not encapsulated by pointwise information. This characteristic cannot be adequately described by the moments of the distribution or by its location and scale parameters alone.

## 5 Experiments

In this section, we demonstrate the empirical efficacy of the proposed framework. First, to verify the estimation procedure for conditional GEV distributions, we evaluate the proposed algorithm in a synthetic 1D scenario. Subsequently, we apply our approach to several semi-synthetic high-dimensional datasets.

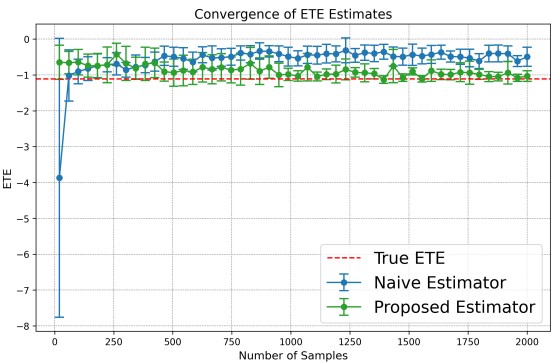

Figure 3: Convergence of different Extreme Treatment Effect (ETE) estimators over increasing sample sizes, showcasing the performance of the Naive Estimator and Proposed Estimator compared to the True ETE. The error bars represent the variability in the estimates across different simulations.

**Baseline Treatment Effect Models.** The proposed framework can be used in conjunction with existing CATE estimation methods. As such, we consider a few models used in the literature and adapt them to the proposed framework for estimating the CETE. The *s*-**learner** method treats the treatment variable as another covariate and builds a neural network $f(x,t)$ to estimate the potential outcomes and then estimates the ITE as $\hat{\tau}(x) = f(x,1) - f(x,0)$. The *t*-**learner** builds two neural networks, one for each treatment group $f_1(x)$ and $f_0(x)$, and the individual outcome is given by $\hat{\tau}(x) = f_1(x) - f_0(x)$ (Künzel et al., 2019). The **TARNet** model is a framework for estimating conditional average treatment effects with counterfactual balancing (Shalit et al., 2017). It consists of a pair of functions, $(\Phi, h)$, where $\Phi$ is a representation learning function and $h$ is a function that learns the two potential outcomes in the representation space. CATE is then estimated as $\hat{\tau}(x) = h(\Phi(x), 1) - h(\Phi(x), 0)$ The TARNet model minimizes the objective $\mathcal{L}(\Phi, h)$, which balances the model's performance on factual data and the similarity of the representations in the latent space using the integral probability metric (IPM). The balancing weight, $\alpha$, controls the trade-off between the similarity and model performance. When $\alpha = 0$ the model is denoted by TARNet, when $\alpha > 0$ the model is known as counterfactual regression (CFR). It corresponds to the 1-Wasserstein distance for 1-Lipschitz functions Villani (2009).

## 5.1 Synthetic Data Experiments

We commence by validating our proposed method through an experiment involving synthetic data. We generate 5000 samples for $Y_0|X = x$ from a Generalized Extreme Value (GEV) distribution, $\text{GEV}(\exp(x), 1, \xi_0(x))$, where $\xi_0(x) = 1 + |x|^{0.1}$, and for $Y_1|X = x$ from $\text{GEV}(x^2, 1, \xi_1(x))$, with $\xi_1(x) = \log(1.1 + x^2)$. The feature variable $X$ is sampled from a normal distribution, $\mathcal{N}(0,1)$, and the propensity score is set to 0.3 for $X > 0$ and 0.7 otherwise. We evaluate the model on data sampled from $\mathcal{N}(0,1)$ and with a propensity score $p(t = 1|x) = \sigma(50x^2 - 5)$, where $\sigma$ denotes the sigmoid function.

The results for this experiment are presented in Figure 2. We observe that the model accurately learns the shape parameter as a function of the covariates, which leads to precise estimation of the Conditional Extreme Treatment Effect (CETE). We then use this trained neural network to estimate the Extreme Treatment Effect (ETE) by sampling from the learned GEV distribution for potential outcomes, since we have learned the shape parameter. We fit a GEV to the marginal samples of the outcomes $\{\hat{z}_t^{(j)}\}_{j=1}^N$. We also evaluate the naive estimator by fitting a GEV to the observed treatment outcomes $\{z_1^{(j)}\}_{j=1}^{N_1}$ and $\{z_0^{(j)}\}_{j=1}^{N_0}$, where $N_1$ and $N_0$ are the cardinalities of the treatment and control groups, respectively. We observe that our estimator exhibits unbiased behavior, unlike the naive estimator. The convergence behavior of both the naive and the proposed ETE estimators are presented in Figure 3.

Table 1: Comparison of the CETE performance of our proposed estimator. The error bars are calculated for various random seeds.

| Model | IHDP (original) | IHDP (Fréchet) | IHDP (Weibull) |
|---|---|---|---|
| S-Learner | $0.048 \pm 0.028$ | $0.396 \pm 0.060$ | $0.307 \pm 0.070$ |
| T-Learner | $0.188 \pm 0.053$ | $0.634 \pm 0.307$ | $0.345 \pm 0.004$ |
| TARNet | $0.119 \pm 0.071$ | $0.223 \pm 0.105$ | $0.278 \pm 0.172$ |
| CFR(Wass) | $0.069 \pm 0.004$ | $0.211 \pm 0.104$ | $0.261 \pm 0.061$ |

## 5.2 Semi-synthetic data experiments

Having validated the proposed method in estimating conditional GEVs, we now proceed to apply this in the causal inference setting to compute the ETE and the CETE.

**Dataset descriptions.** We present a representative family of semi-synthetic datasets tailored for estimating the CETE. Specifically, we use the Infant Health and Development Program (IHDP) dataset (Ramey et al., 1992) and two variants corresponding to different MDAs. Both datasets consider potential outcomes generated based on known functions of the observed covariates. The selection bias is the same in the three datasets, provided by the real-world experiment, and we produce the potential outcomes following different GEVs. We provide more detailed descriptions of these datasets in Appendix F.

**CETE results.** We present the performance for CETE estimation in Table 1. The results suggest that the CFR(Wass) model performs the best for the Fréchet and the Weibull cases while the SNet performs best for the IHDP original case (the Gumbel case).

**ETE results.** We consider the performance of our proposed ETE estimator compared to that of a naive estimator, which is based on the block maxima method applied directly to the potential outcomes. The naive estimator is then given by: $\tau_{\text{naive}} = \underline{\xi}_{1,\text{naive}} - \underline{\xi}_{0,\text{naive}}$, where $\xi_{t,\text{naive}}$ correspond to the estimated shape parameter for the MDA of the conditional outcome $Y|T = t$. It is important to note that the conditional outcome $Y|T = t$ is distinct from the potential outcome $Y_t$. The ETE results are summarized in Table 2

Table 2: ETE Performance comparison under the naive and proposed estimators across different datasets using the CFR(Wass) model.

| Dataset | $\varepsilon_{\text{ETE}}(\text{Naive})$ | $\varepsilon_{\text{ETE}}(\text{Ours})$ |
|---|---|---|
| IHDP (ORIGINAL) | 0.094 | 0.001 |
| IHDP (FRÉCHET) | 0.346 | 0.095 |
| IHDP (WEIBULL) | 0.931 | 0.827 |

## 6 Conclusion

In this work, we propose a new framework for estimating extreme treatment effects. We define new statistical quantities CETE and ETE capturing extreme treatment effects. We describe conditions to guarantee the identifiability and the consistency of the proposed estimators. To circumvent the lack of tail data, we propose a practical approach to sample from the MDAs and introduce a neural network approach for learning the GEV parameters. Our contributions offer new insights into the estimation of extreme treatment effects and provide a basis for further research in this area. Additionally, further analysis in developing a computational method leveraging the point process perspective should be considered.

**Limitations** The main limitation of the method is we require regularity on the space of conditional outcomes to be able to use our block maxima surrogate. Circumventing this requires multiple outcomes per individual,

which is often impossible in practice. Understanding any additional structure that can be used for the block maxima is a direction for future research to address this limitation.

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

# Appendix

## A  Simulating GEV Potential Outcomes

In this illustration, we model potential outcomes under control and treatment scenarios for a population of $n = 5$ individuals. Each individual $k \in \{1, \dots, n\}$ is associated with two potential outcomes. For the control scenario, the potential outcome for individual $k$ follows a Generalized Extreme Value (GEV) distribution with a location parameter $\mu_k^0 = 5 + k$, a scale parameter $\sigma_k^0 = 1$, and a shape parameter $\xi_k^0 = 0.5$. For the treatment scenario, the potential outcome for individual $k$ follows a GEV distribution with a location parameter $\mu_k^1 = 6 + k$, a scale parameter $\sigma_k^1 = 1.5$, and a shape parameter $\xi_k^1 = 1.5 + 0.1k$.

Figure 4 illustrates the distribution of the potential outcomes for the $n = 5$ individuals, each with distinct location and shape parameters. In our work, we study the differences in the $\xi$ parameter, which, as the figure shows, dictates how the tail of the distribution behaves.

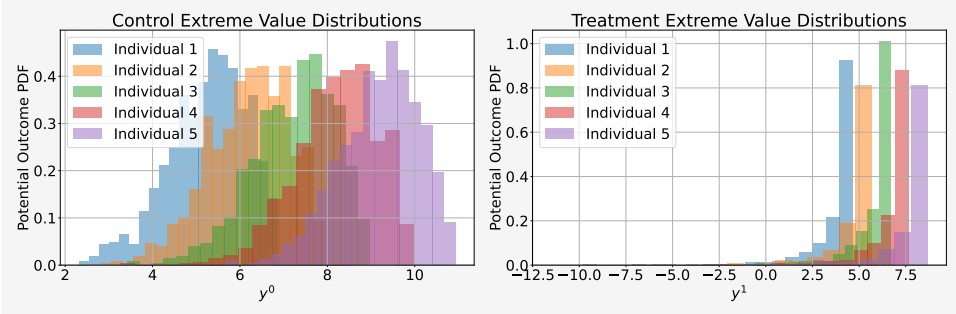

Figure 4: Different conditional outcomes for different treatment groups. The goal of our method is to estimate the difference in the shape parameters of these limiting GEVs.

## B  Theoretical Results

In this section, we begin by providing proofs for the remarks presented in the paper, which support the proposed ETE and CETE definitions. This is followed by a discussion of other potential definitions for the extreme treatment effects. Subsequently, we present proofs for the causal inference results presented in the main text. Finally, we provide a theoretical analysis of the $\varepsilon$-max sampler algorithm utilized in our empirical settings.

### B.1  Proof of Regularly Varying Remark

**Remark 3.3.** *Let $\xi_0, \xi_1 > 0$. Let $S_1(Y_1), S_0(Y_0)$ be the survival distributions of the potential outcome variables with positive support for the treatment and control groups respectively. Then if $\tau_{ext} > 0$, then $S_0(Y_0)/S_1(Y_1) \in RV(\xi_1 - \xi_0)$. Moreover, $S_1(Y_1) < S_0(Y_0)$ for $Y_t \gg 1$.*

*Proof.* Define $F_0, F_1$ as the CDFs of the outcome variables for the control and treatment groups respectively. Then $S_0 = 1 - F_0$ and $S_1 = 1 - F_1$. $F_0, F_1$ are also assumed to be regularly varying with indices of regular variation as $\xi_0, \xi_1$. By the Karamata characterization theorem, we can write $F_0, F_1$ in terms of a slowly-varying function, that is, a function where $\xi = 0$. We will call this function $L(Y)$ and then write $S_0 = Y^{-\xi_0} L_0(Y)$ and $S_1 = Y^{-\xi_1} L_1(Y)$. As $Y \to \infty$, the polynomial component dominates, and we can compare the ratio of the polynomial components. Taking the ratio, we get

$$\lim_{Y \to \infty} \frac{S_0}{S_1} \sim \frac{Y^{-\xi_0}}{Y^{-\xi_1}} = Y^{\xi_1 - \xi_0}.$$

Thus, the difference in the tail indices can be seen as the ratio of the survival functions for regions deep in the tail. □

## B.2 Proof of Point Process Remark

**Remark 3.4** (Expected Number of Events). *Consider a causal inference task with potential outcomes variable $Y_t$, $t \in \{0, 1\}$, that $Y_t$ is in the maximum domain of attraction of a GEV with rate $\xi$, and ETE given by $\tau_{ext} > 0$. Suppose that $\mathcal{R} \subset \mathbb{R}$ is the Borel set of interest that should be quantified for the causal inference task with $|\inf \mathcal{R}| \gg 0$. Then, $\mathbb{E}[N_{Y_1}(\mathcal{R})] > \mathbb{E}[N_{Y_0}(\mathcal{R})]$ where $N_{Y_t}(\mathcal{R}) = \sum_{i=1}^{\infty} \delta_{Y_t^{(i)} \in \mathcal{R}}$ for $t \in \{0, 1\}$ is a counting process over outcomes on some Borel subset of $\mathbb{R}$.*

*Proof.* From our assumptions on $\mathcal{R}$, we suppose that the GEV is the appropriate description of the events. Additionally, suppose that both $Y_0$ and $Y_1$ have the same limiting sequence $a_n, b_n$ and by EVT

$$\mathbb{E}[N_{Y_0}(\mathcal{R})] = [1 + \xi_0 c_{\mathcal{R}}]^{-1/\xi_0}$$

and

$$\mathbb{E}[N_{Y_1}(\mathcal{R})] = [1 + \xi_1 c_{\mathcal{R}}]^{-1/\xi_1}.$$

Since $\xi_1 > \xi_0$,

$$1 + \xi_1 c_{\mathcal{R}} > 1 + \xi_0 c_{\mathcal{R}}$$
$$(1 + \xi_1 c_{\mathcal{R}})^{1/\xi_1} > (1 + \xi_0 c_{\mathcal{R}})^{1/\xi_1}$$
$$(1 + \xi_1 c_{\mathcal{R}})^{1/\xi_1} < (1 + \xi_0 c_{\mathcal{R}})^{1/\xi_0}$$
$$(1 + \xi_1 c_{\mathcal{R}})^{-1/\xi_1} > (1 + \xi_0 c_{\mathcal{R}})^{-1/\xi_0}$$

with $c_{\mathcal{R}} \in \mathbb{R}_+ \cup \{0\}$. □

## B.3 Possible Definitions for Extreme Treatment Effects

In this section, we note that while we motivate our definition for the extreme treatment effect from an intuitive perspective, a point process perspective, and from a regular variation perspective, it is possible to define other functions of the shape parameter $\Psi(\xi_1(x), \xi_0(x))$ that capture other properties. Our presented methods can be extended to these general functions, as the main problem in causal inference lies in estimating $\xi_1(x)$ and $\xi_0(x)$. Moreover, the choice of the difference between the shape parameters guarantees that the statistical properties of the estimators of shape parameters (e.g., expectation and variance) can be easily extended to the estimator of the extreme treatment effects, which may not be the case for other functions such as the ratio of the shape parameters. It is important to highlight that, at times, quantifying the proportional change of the shape parameter can be essential (thus, taking the ratio would be more justified). In contrast, in some situations, the magnitude of change itself is significant regardless of the initial value (for instance, if having a heavier tail above 2 is critical and the control shape parameter is 0, taking the difference provides more information). Therefore, different definitions might be more relevant depending on the context

## B.4 Causal Inference Proofs

We restate the main statements and then state the proofs.

**Proposition 3.6.** *If unconfoundedness holds then tail unconfoundedness holds.*

*Proof.* Assuming that unconfoundedness holds, and letting $t \in \mathcal{T}$, we have

$$Y_t \perp\!\!\!\perp T \mid X$$

Therefore,

$$P(Y_t \mid X, T) = P(Y_t \mid X).$$

For $\{Y^{(k)}\}_{k=1}^{m}$ i.i.d copies of $Y$, we have that

$$\forall k \in \{1, \ldots, m\}, \ P(Y_t^{(k)} \mid X, T) = P(Y_t^{(k)} \mid X)$$

By taking the limit on both sides, we have

$$\lim_{m \to \infty} P(Z_t^m \mid X, T) = \lim_{m \to \infty} P(Z_t^m \mid X)$$

which implies tail unconfoundedness holds. □

**Proposition 3.9** (Identifiability). *Under Assumptions 2.2, 2.3, and 3.5, the CETE and ETE estimators defined in Definition 3.7 and Definition 3.8 respectively are causally identifiable. Moreover, for $t \in \{0, 1\}$, the proposed estimator satisfies*

$$\hat{\xi}_t = \arg\max_{\theta \in \Theta} \sum_{j=1}^{N} \log p \left( Z^{(j)} \mid g_\theta(x^{(j)}), t^{(j)} \right) \mathbb{1}_{t^{(j)}=t}.$$

*Proof.* By definition, we have

$$\hat{\theta}_t = \arg\max_{\theta} \lim_{m \to \infty} \sum_{j=1}^{N} \log P \left( Z_t^m(x^{(j)}) \mid \theta \left( x^{(j)} \right) \right)$$

$$= \arg\max_{\theta} \sum_{j=1}^{N} \lim_{m \to \infty} \log P \left( Z_t^m(x^{(j)}) \mid \theta \left( x^{(j)} \right) \right)$$

By tail unconfoundedness, we have

$$\lim_{m \to \infty} P \left( Z_t^m(x^{(j)}) \mid \theta \left( x^{(j)} \right) \right) = \lim_{m \to \infty} P \left( Z_t^m(x^{(j)}) \mid \theta \left( x^{(j)} \right), T = t \right)$$

From the consistency assumption, it follows that

$$\lim_{m \to \infty} P \left( Z_t^m(x^{(j)}) \mid \theta \left( x^{(j)} \right), T = t \right) = \lim_{m \to \infty} P \left( Z^m(x^{(j)}) \mid \theta \left( x^{(j)} \right), T = t \right)$$

Notice that the right-hand-side quantity does not involve any potential outcomes. We then have

$$\lim_{m \to \infty} P \left( Z_t^m(x^{(j)}) \mid \theta \left( x^{(j)} \right) \right) = \lim_{m \to \infty} P \left( Z^m(x^{(j)}) \mid \theta \left( x^{(j)} \right), T = t \right)$$

Hence, the estimator is identifiable. □

**Remark 3.10.** *The ETE estimator defined in Definition 3.8 is consistent and asymptotically normal.*

*Proof.* Given that the MLE estimator for the shape parameters is consistent, we can apply continuous mapping theorem to deduce that the CETE estimator is also consistent, for a fixed individual as his potential outcomes observations go to infinity. Additionally, because the MLE estimator for the shape parameters is asymptotically normal, we can use the Delta method to conclude that the ETE estimator is asymptotically normal. □

## C Theoretical Analysis for $\varepsilon$-max-sampler

The main idea of this approach is to consider a ball around each individual outcome and then use the maxima of all outcomes within this ball as an estimate of data in the tails. We prove that this converges to the true maxima under suitable regularity conditions. For treatment groups $t \in \{0, 1\}$, the conditional potential outcome variable $\{(Y|X = x) = Y_t(x), \ x \in \mathcal{X}\}$ forms a stochastic process over $\mathcal{X}$. For every $x \in \mathcal{X}$, we assume that $\left( Y_t^{(1)}(x), Y_t^{(2)}(x), \ldots, Y_i^{(m)}(x) \right)$ are iid random variables with the same distribution as $Y(x)$.

We define the max-sampler: $Z_t^m(x) = \max_{1 \le k \le m} Y_t^{(k)}(x)$, $x \in \mathcal{X}$. Let $\varepsilon > 0$, and let $B_\varepsilon(x) = \{x' \mid x' \in \mathbb{R}^d$ and $\|x - x'\| < \varepsilon\}$ be the open ball around $x$ with radius $\varepsilon$. Let $B_\varepsilon^m(x)$ be a subset of $m$ points of $B_\varepsilon(x)$. We define the $\varepsilon$-max-sampler as

$$\hat{Z}_{t,\epsilon}^m(x) = \max_{x' \in B_\epsilon^m(x)} Y_t(x') \tag{2}$$

We next study the theoretical relationships between $Z_t^m(x)$ and $\hat{Z}_{t,\varepsilon}^m(x)$ and determine the conditions under which $\hat{Z}_{t,\varepsilon}^m(x)$ converges to $Z_t^m(x)$.

**Proposition C.1** (Convergence of $\varepsilon$-max-sampler)**.** *Let $t \in \{0,1\}$ and $m \in \mathbb{N}$. Assume that for every $x, x' \in \mathcal{X}$, such that $x \ne x'$, we have that $Y(x) \perp\!\!\!\perp Y(x')$ and $x \mapsto P(Y(x) \le y)$ is a continuous function of $x$. Then for every limit point, $x \in \mathcal{X}$, the following convergence holds in distribution*

$$\forall x \in \mathcal{X}, \ \hat{Z}_{t,\varepsilon}^m(x) \xrightarrow[\varepsilon \to 0]{} Z_t^m(x)$$

*Proof.* Denote by $F(\cdot \mid x)$ the CDF of $\hat{Z}_{t,\varepsilon}^m(x)$ and let $x_1, \ldots, x_n$ be $n$ elements of the $\varepsilon$-ball. We have that

$$
\begin{aligned}
F(z|x) &= P(\hat{Z}_{t,\varepsilon}^m(x) \le z) \\
&= P(Y(x_1) \le z, \ldots, Y(x_n) \le z) \\
&= \Pi_{k=1}^n P(Y(x_k) \le z)
\end{aligned}
$$

we have that when $\epsilon \to 0$ all the points in the $\varepsilon$−ball $x_k \to x$, therefore,

$$\Pi_{k=1}^n P(Y(x_k) \le z) \to \Pi_{k=1}^n P(Y^{(k)}(x) \le z), \text{ as } \varepsilon \to 0.$$

We then have that

$$\Pi_{k=1}^n P(Y^{(k)}(x) \le z) = P(Y^{(1)}(x) \le z, \ldots, Y^{(n)}(x) \le z)$$

and the right-hand side corresponds to the CDF of $Z_t^m(x)$. $\square$

We note that the first independence assumption is often referred to as the no interference assumption between the units (see also the stable unit treatment value assumption (Imbens & Rubin, 2015; Keele, 2015)). From this, we can bound the difference between the $\varepsilon$-max-sampler estimate and the true potential outcome under a Lipschitz assumption on the potential outcomes:

**Proposition C.2.** *Let $t \in \{0,1\}$. Assume that the potential outcomes are positive and that $x \to Y_t(x)$ is $K$-Lipschitz almost surely, then*

$$\forall n \in \mathbb{N}, \forall \varepsilon > 0, \ \left\| \hat{Z}_{t,\varepsilon}^m(x) - Z_t^m(x) \right\| \le K\varepsilon \ \ a.s.$$

The theorem states that the error grows at most at the rate of the Lipschitz constant.

*Proof.* Let $n \in \mathbb{N}$ and let $\varepsilon > 0$. We have that

$$\forall x, x' \in \mathcal{X}, \ \|Y(x) - Y(x')\| \le K \|x - x'\|$$

For the $\varepsilon$−ball around $x$ we have that

$$\forall x' \in B_\varepsilon(x), \ \|Y(x) - Y(x')\| \le K \|x - x'\|.$$

By taking the max over the right-hand side, we have that

$$\forall x' \in B_\varepsilon(x), \ \|Y(x) - Y(x')\| \le K\varepsilon.$$

Since the potential outcomes are positive we have for $x_1, \ldots, x_m \in B_\varepsilon(x)$

$$\left\| \max_{1 \le k \le m} Y(x_k) - \max_{1 \le k \le m} Y^k(x) \right\| \le \max_{1 \le k \le m} \left\| Y(x_k) - Y^k(x) \right\|.$$

Therefore,

$$\left\| \hat{Z}_{t,\varepsilon}^m(x) - Z_t^m(x) \right\| \le K\varepsilon.$$

$\square$

## D  Tail Unconfoundednes Examples

In this section, we present two quantitative examples where tail unconfoundedness is satisfied but unconfoundedness does not hold.

**Example 1:**  Let the mass function $P(Y_1 \mid T = t, X)$ be a Bernoulli distribution with a non-binary treatment parameter $t \in (0, 1)$. To compute the probability that the maximum is equal to 1, we write

$$P(Z_1^n = 1 \mid T = t, X) = 1 - P(Y_1^1 = 0, \ldots, Y_1^n = 0 \mid T = t, X)$$
$$= 1 - P^n(Y_1 \mid T = t, X)$$
$$= 1 - t^n.$$

This is independent of $t$ as $n \to \infty$ (since $t \in (0, 1)$).

**Example 2:**  Consider the density function $p(y_1 \mid x, t)$ that follows a Beta distribution $\mathcal{B}(\alpha, \beta)$ with parameters $\alpha = x$ and $\beta = t + 1$ for $x, t > 0$. The domain of attraction of this distribution is GEV$(0, 1, \xi(x))$ with $\xi(x) = -x^{-1}$ with the following normalizing constants Roncalli (2020):

$$b_m(x) = 1 \quad \text{and} \quad a_m(x) = \left( \frac{m\Gamma(x + t + 1)}{\Gamma(x)\Gamma(t + 1)} \right)^{-1/(t+1)}.$$

We see that as $m \to \infty$, $a_m \to 0$. Then, the limiting distribution of $Z_1^n$ only depends on $x$ and does not depend on $t$, implying that the tail unconfoundedness holds.

## E  Algorithm

We describe the algorithm for the CETE estimation in algorithm 1.

## F  Datasets Descriptions

In this section, we present a detailed description of the datasets used in the numerical study. A summary is provided in Table 3.

**IHDP**  The IHDP dataset was first introduced by Hill (2011) and it is based on real covariates available from the Infant Health and Development Program (IHDP), which studied the effect of development programs on children. The features in this dataset come from a randomized controlled trial. The potential outcomes are simulated using Setting B. The dataset consists of 747 individuals (specifically, 139 in the treatment group and 608 in the control group), each with 25 features. The potential outcomes are generated according to the following distributions for the control group:

$$Y_0 = \exp(\beta^T \cdot (X + W)) + \eta_0, \quad \text{with } \eta_0 \sim \mathcal{N}(0, 1)$$

and for the treatment group:

$$Y_1 = \beta^T (X + W) - \omega + \eta_1, \quad \text{with } \eta_1 \sim \mathcal{N}(0, 1)$$

where $W$ has the same dimension as $X$ with all entries equal 0.5 and $\omega = 4$. The regression 25 dimensional coefficient vector $\beta$ is randomly sampled from a categorical distribution with the support $(0, 0.1, 0.2, 0.3, 0.4)$ and the respective probabilities $\mu = (0.6, 0.1, 0.1, 0.1, 0.1)$. We refer to this dataset as IHDP (Original), which corresponds to a Gumbel domain of attraction. We also generate two new versions of IHDP by changing the noise variables of the potential outcomes. The first version is given by:

$$Y_0 = \exp(\beta^T \cdot (X + W)) + \eta_0, \quad \text{with } \eta_0 \sim \text{GEV}(0, 1, 0.1 + \|X\|_2^{1/10})$$

and

$$Y_1 = \beta^T (X + W) - \omega + \eta_1, \quad \text{with } \eta_1 \sim \text{GEV}(0, 1, \log(1.1 + \|X\|_2))$$

---

**Algorithm 1:** Learning the Parameters of the Domain of Attraction of Individual Potential Outcomes

---

**Data:** $S = \{(x^{(i)}, t^{(i)}, y^{(i)})\}_{i=1}^n$

**Input:** A neural network $N_\theta = \{(\hat{\mu}_t, \hat{\sigma}_t, \hat{\xi}_t)\}_{t=0}^1$ : $s$-Learner, $t$-Learner, TARNet, or CFR

**Output:** A trained neural network $N_{\theta*}$

**1 Function** `MaxSampler(S, m):`

**2**     Initialize an empty set $S^m$

**3**     Number of clusters:

**4**     $K = n \text{ div } m$

**5**     Run $K$-means to and cluster the data into $K$ clusters.

**6**     **for** $i = 1, 2, \ldots, K$ **do**

**7**        Take $S_i = \{j \mid (x^{(j)}, t^{(j)}, y^{(j)}) \text{ in the } i^{th} \text{ cluster}\}$

**8**        Select $s = \arg\max_{j \in S_i} y^{(j)}$

**9**        Add $(x^{(s)}, t^{(s)}, y^{(s)})$ to $S^m$

**10**     **return** $S^m$

**11 Function** `Loss(`$S^m, N_\theta$`):`

**12**     For each $(x^{(i)}, t^{(i)}, y^{(i)}) \in S^m$ the negative conditional log-likelihood is

**13**     $L_i = -\log(P(y^{(i)} \mid \mu_0(x^{(i)}), \sigma_0(x^{(i)}), \xi_0(x^{(i)}))) \mathbf{1}_{t^{(i)}=0}$

       $- \log(P(y^{(i)} \mid \mu_1(x^{(i)}), \sigma_1(x^{(i)}), \xi_1(x^{(i)}))) \mathbf{1}_{t^{(i)}=1}$

**14**     **return** $\frac{1}{n} \sum_{i=1}^n L_i$

**15 Function** `Main:`

**16**     Choose $m$

**17**     Run the max-sampler

**18**     $S^m = $ `MaxSampler(`$S, m$`)`

**19**     Train the neural network $N_\theta$ by minimizing the loss:

**20**     L = `Loss(`$S^m, N_\theta$`)`

**21**     **return** The trained neural network $N_{\theta*}$

---

Table 3: Overview of the causal inference datasets for CETE and ETE Estimation.

| Name | Type | IPO MDA |
|---|---|---|
| IHDP (original) | Semi-Synthetic | Gumbel |
| IHDP (Fréchet) | Semi-Synthetic | Fréchet |
| IHDP (Weibull) | Semi-Synthetic | Weibull |

Individual Potential Outcomes Maximum Domain of Attraction (**IPO MDA**) if consistent across all individuals.

We refer to this version as IHDP (Fréchet). The second version is generated with the following potential outcomes:

$$Y_0 = \exp(\beta^T \cdot (X + W)) + \eta_0, \text{ with } \eta_0 \sim \text{GEV}(0, 1, -0.1 - \|X\|_2^{1/10})$$

and

$$Y_1 = \beta^T(X + W) - \omega + \eta_1, \text{ with } \eta_1 \sim \text{GEV}(0, 1, -\log(1.1 + \|X\|_2))$$

We refer to this version as IHDP (Weibull).

## G GEV Likelihoods

For reference, we provide the correspondences between the bulk distributions and tail distributions for our simulated experiments.

### G.0.1 Gumbel Distribution

The density function of a Gumbel is

$$f(y; \mu(x), \beta(x)) = \frac{1}{\beta(x)} \exp\left(-\left(\frac{y - \mu(x)}{\beta(x)} + e^{-\frac{y - \mu(x)}{\beta(x)}}\right)\right)$$

with $\mu(x)$ a real function for the location parameter and $\beta(x)$ a positive real function of the scale.

Then, the conditional negative log-likelihood is

$$L(\mu, \beta) = \sum_{i=1}^{n} \left(\frac{y_i - \mu(x_i)}{\beta(x_i)} + e^{\frac{y_i - \mu(x_i)}{\beta(x_i)}}\right) + n \log \beta(x_i).$$

### G.0.2 Fréchet Distribution

The density function of a Fréchet distribution is

$$f(y; \alpha(x), s(x), m(x)) = \frac{\alpha(x)}{s(x)} \left(\frac{y - m(x)}{s(x)}\right)^{-1 - \alpha(x)} e^{-\left(\frac{y - m(x)}{s(x)}\right)^{-\alpha(x)}}.$$

Then, the conditional negative log-likelihood is

$$L(\alpha, s, m) = \sum_{i=1}^{n} \left[(1 + \alpha(x_i)) \log\left(\frac{y_i - m(x_i)}{s(x_i)}\right) + \left(\frac{y_i - m(x_i)}{s(x_i)}\right)^{-\alpha(x_i)} + \log\left(\frac{s(x_i)}{\alpha(x_i)}\right)\right].$$

### G.0.3 Weibull Distribution

The density function of a Weibull is

$$f(y; \lambda(x), k(x)) = \frac{k(x)}{\lambda(x)} \left(\frac{y}{\lambda(x)}\right)^{k(x) - 1} \exp\left(-(y/\lambda(x))^{k(x)}\right) \mathbb{1}_{y \geq 0}$$

with $k(x)$ and $\lambda(x)$ two positive functions of the shape and the scale, respectively. The conditional negative log-likelihood is

$$L(\lambda, k) = \sum_{i=1}^{n} \left(\frac{y_i}{\lambda(x_i)}\right)^{k(x_i)} + k(x_i) \log(\lambda(x_i)) - \log(k(x_i)) + (1 - k(x_i)) \log(y_i).$$

## H Convergence to GEV Distributions

### H.0.1 Gaussian to Gumbel

Let $Y(x) \sim \mathcal{N}(\mu(x), \sigma(x))$. The limiting GEV is given by

$$\frac{Z_m(x) - b_m(x)}{a_m(x)} \xrightarrow[m \to +\infty]{} \text{Gumbel}(0, 1)$$

where

$$b_m = \mu(x) + \sigma(x) \left(\sqrt{2 \log(m)} - \frac{\log(\log(m)) + \log(4\pi)}{2\sqrt{\log(m)}}\right)$$

and

$$a_m = \frac{\sigma(x)}{\sqrt{2 \log(m)}}.$$

This implies that for large enough $m$,

$$Z_m(x) \approx c_m(x)g + d_m(x)$$

with $g \sim \text{Gumbel}(0, 1)$.

### H.0.2 Log-Gamma to Fréchet

Let $Y(x) \sim \mathcal{LG}(\alpha(x), \beta(x))$. The limiting GEV is given by

$$\frac{Z_m(x) - b_m(x)}{a_m(x)} \underset{m \to +\infty}{\longrightarrow} \text{GEV}(0, 1, \beta(x)^{-1})$$

where

$$a_m(x) = 0 \text{ and } b_m(x) = \frac{\left(m \log(m)^{\alpha(x)-1}\right)^{1/\beta(x)}}{\Gamma(\alpha(x))}.$$

Then,

$$Z_m(x) \approx a_m(x) \, \text{GEV}(0, 1, \xi(x)) + b_m(x)$$

with

$$\xi(x) = \beta(x)^{-1}.$$

### H.0.3 Beta to Weibull

Let $Y(x) \sim \mathcal{B}(\alpha(x), \beta(x))$. We have that

$$\frac{Z_m(x) - b_m(x)}{a_m(x)} \underset{m \to +\infty}{\longrightarrow} \text{GEV}(0, 1, -\alpha(x)^{-1})$$

where

$$b_m(x) = 1 \quad \text{and} \quad a_m(x) = \left(\frac{m\Gamma(\alpha(x) + \beta(x))}{\Gamma(\alpha(x))\Gamma(\beta(x) + 1)}\right)^{-1/\beta(x)}.$$

Then,

$$Z_m(x) \approx a_m(x) \, \text{GEV}(0, 1, \xi(x)) + b_m(x)$$

with

$$\xi(x) = -\alpha(x)^{-1}.$$

## I  Additional Experiments

In this section, we provide additional experiments to validate the proposed method. We start with verifying our methods for estimating conditional GEVs and the estimated scale parameter of MDA for the Log-Gamma distribution. Then, an ablation is performed on the number of the data size $N$ and the dimensionality $D$. Finally, we present an ablation on the number of clusters $K$.

### I.0.1 Estimation from GEV observations

We estimate the functional parameters of the GEV given a dataset $S = \{(x^{(i)}, y^{(i)})\}_{i=1}^{n}$ where $P(Y \mid X = x)$ follows a GEV distribution. We consider the cases of the Weibull, Gumbel, and Fréchet distributions by learning the parameters of each of the distributions. The full likelihood functions for each case were detailed in Appendix G. A comparison of the estimated and true functions is given in Figure 7.

**Conditional Gumbel**  We assume that the data are generated according to $Y(x) \sim \text{Gumbel}(\mu(x), \beta(x))$ with $\mu(x) = x^2$ and $\beta(x) = 0.25 + \frac{|x|}{2}$. Figure 7a shows the performance of our model on this dataset.

**Conditional Weibull**  We assume that the data are generated according to $Y(x) \sim \text{Weibull}(\lambda(x), k(x))$ with $\lambda(x) = 1 + x^2$ and $\beta(x) = 1 + |x|$. Figure 7b illustrates the results of fitting the estimator to the conditional data.

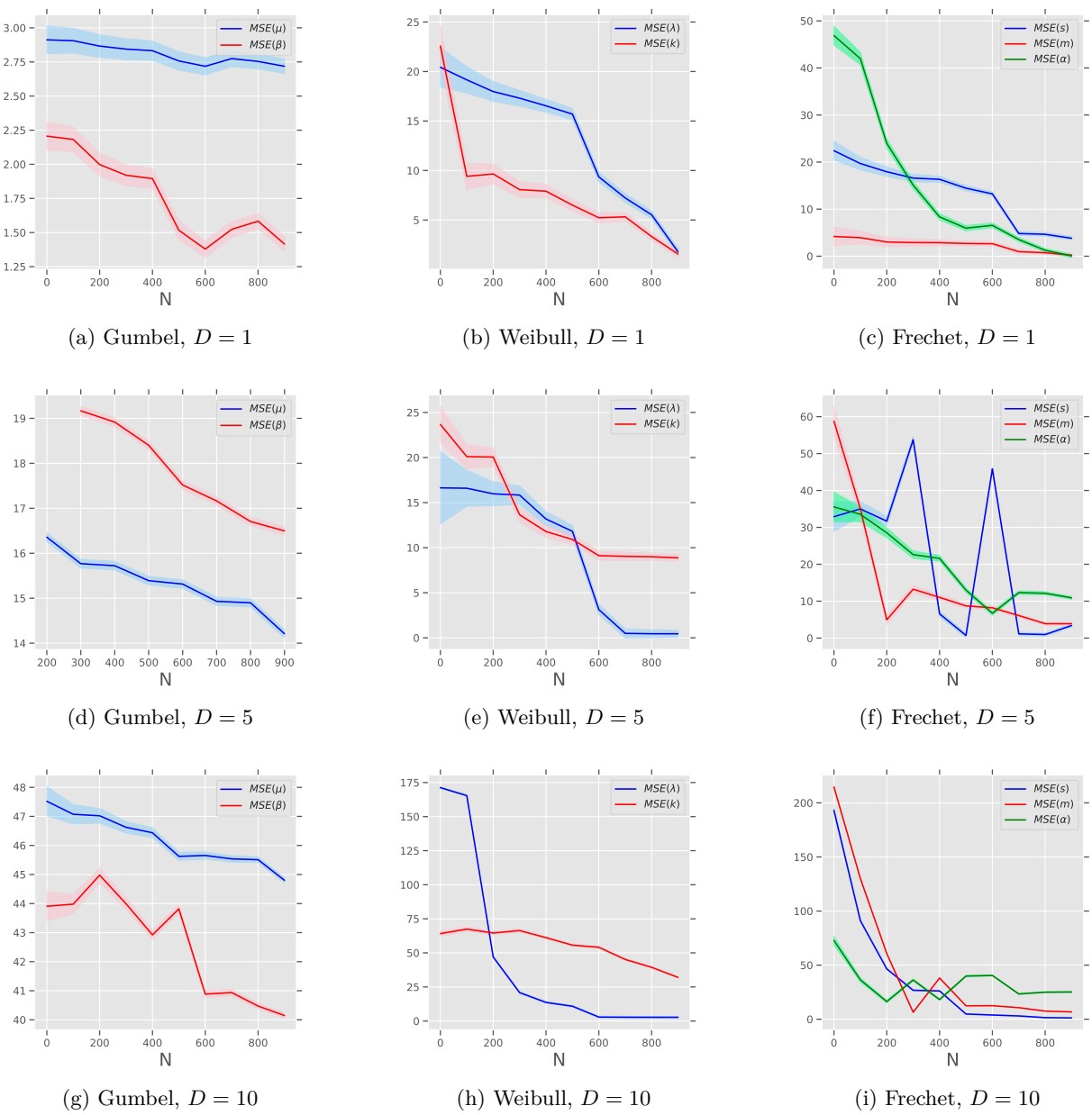

Figure 5: Experiments For Conditional GEV Performance as a function of $N$ and $D$

**Conditional Fréchet** We assume that the data are generated according to: $Y(x) \sim$ Fréchet$(\alpha(x), s(x), m(x))$ where $\alpha(x) = 1 + x^2$, $s(x) = 1 + |x|$, and $m(x) = \exp(x)$. Figure 7c illustrates the results for fitting in the Fréchet case.

These experiments validate the estimation procedure in the main text when observing the extreme data. Next, we consider the case where we observe samples of the data and need to use the $\varepsilon$-max-sampler to estimate the extreme observations. We do this by applying the $\varepsilon$-max-sampler to data from a log-normal distribution and estimate the corresponding Frechét distribution. The results are presented in Figure 9 for estimating the scale $\sigma$ parameter.

We also estimated the scale parameter for the Log-Gamma distribution in Figure 9.

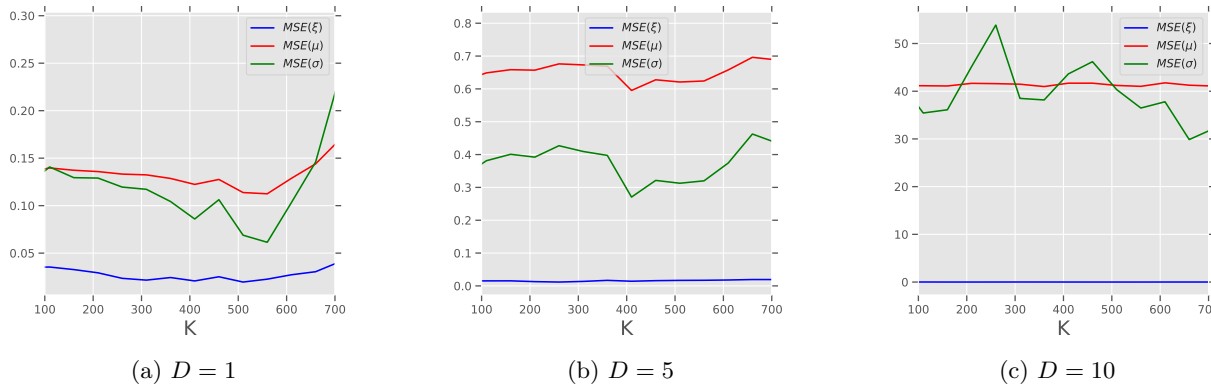

(a) $D = 1$      (b) $D = 5$      (c) $D = 10$

Figure 6: Experiments for the Gaussian to Gumbel case as a function of K

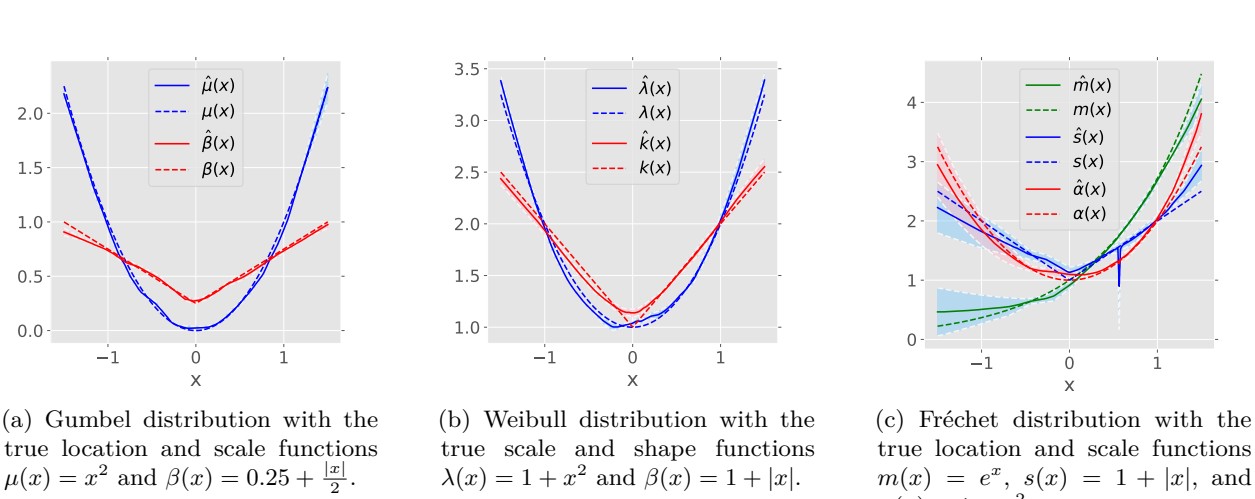

(a) Gumbel distribution with the true location and scale functions $\mu(x) = x^2$ and $\beta(x) = 0.25 + \frac{|x|}{2}$.

(b) Weibull distribution with the true scale and shape functions $\lambda(x) = 1 + x^2$ and $\beta(x) = 1 + |x|$.

(c) Fréchet distribution with the true location and scale functions $m(x) = e^x$, $s(x) = 1 + |x|$, and $\alpha(x) = 1 + x^2$.

Figure 7: Estimation of distribution parameters from GEV observations for Gumbel, Weibull, and Fréchet distributions.

### I.0.2 Estimating conditional limiting GEV distributions

For these experiments, we consider the $\varepsilon$-max-sampler with a total of 2000 random samples of the limiting GEV. In Appendix I, we include detailed experiments on fitting conditional GEV distributions when they are directly observed. We generate data based on a distribution with a known MDA, and then compare the recovered GEV distribution parameters to sequences known for their convergence to the true parameters. We provide full details of the convergence of these distributions to their corresponding GEV distributions in Appendix H. The results are presented in Figure 8 where we compare the true functions to the estimated ones for the different datasets. We additionally provide an ablation study on the impact of varying the block sizes and the dimensionality of the data in Appendix I.

**Gaussian distribution: Gumbel MDA.** We generate $X \sim \mathcal{N}(0,1)$, and $Y(x) \sim \mathcal{N}(\mu(x), \sigma(x))$, where $\mu(x) = |x|$ and $\sigma(x) = x^2$. The corresponding GEV is the Gumbel distribution. Moreover, we estimate the limiting conditional GEV parameters with $\hat{\mu}(x), \hat{\sigma}(x), \hat{\xi}(x)$. Figure 8a illustrates this result where we compare a sequence converging to the ground truth to the estimated parameters.

**Beta distribution: Weibull MDA.** We generate samples $X \sim \mathcal{N}(0,1)$ and $Y(x) \sim \mathcal{B}(\alpha(x), \beta(x))$ with $\alpha(x) = |x| + 1$ and $\beta(x) = x^2 + 1$. The corresponding GEV is a Weibull distribution. We illustrate the estimates of these parameters in Figure 8b.

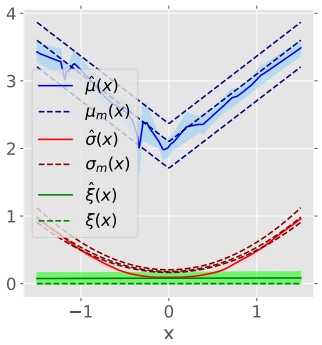
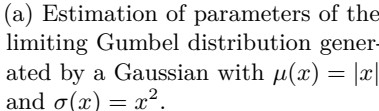
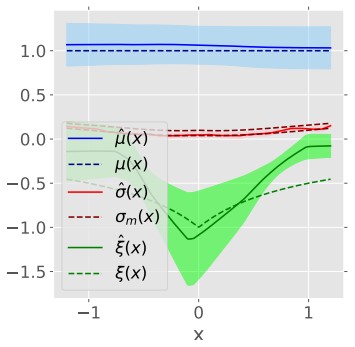
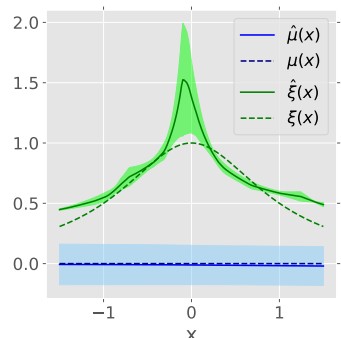

(a) Estimation of parameters of the limiting Gumbel distribution generated by a Gaussian with $\mu(x) = |x|$ and $\sigma(x) = x^2$.

(b) Estimation of parameters of limiting Weibull generated by a beta distribution with the parameters given by $\alpha(x) = |x| + 1$ and $\beta(x) = x^2 + 1$.

(c) Estimation of parameters of limiting Fréchet distribution generated by a log-gamma distribution with parameters given by $\alpha(x) = |x| + 1$ and $\beta(x) = x^2 + 1$.

Figure 8: Estimation of limiting GEV distribution parameters from observations of Gaussian, beta, and log-gamma distributions. All parameters are estimated by taking the max of the $K$-means partition of the feature space.

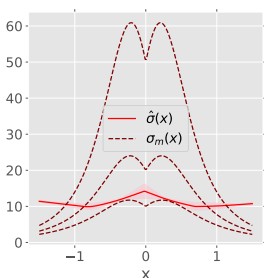

Figure 9: Estimation of the limiting scale parameter $\sigma$ of the maximum of the log-gamma distribution.

**Log-Gamma distribution: Fréchet MDA.** We generate samples $X \sim \mathcal{N}(0, 1)$ and $Y(x) \sim \mathcal{LG}(\alpha(x), \beta(x))$ with $\alpha(x) = |x| + 1$ and $\beta(x) = x^2 + 1$ where $\mathcal{LG}$ is the log-gamma distribution. The corresponding GEV is a Fréchet distribution. Figure 8c illustrates the performance of the proposed method. The results for the scale parameter are presented in Appendix I

**Ablation on $N$ and $D$** We study how the performance of our method changes when varying the data size $N$ and the dimensionality $D$, as illustrated in Figure 5. The ablation suggests that the Weibull and Frechet cases scale when observing large $N$ but the Gumbel case is more difficult to estimate.

**Ablation on number of clusters** We study the performance of our method when varying the number of clusters $K$ and the dimensionality $D$ for a fixed number of data points $N = 10^4$. The results are illustrated in Figure 6.

