# OpenReview forum: "Treatment Effects in Extreme Regimes"
_TMLR — Under review for TMLR_

### Review · Reviewer_FAD9 · 2026-06-17

**Summary Of Contributions:**

The paper studies **treatment effects in the tails** of the potential-outcome distributions rather than at the mean. Building on Extreme Value Theory (EVT), it models each potential outcome's tail with a Generalized Extreme Value (GEV) law and defines the treatment effect as the **difference in GEV shape parameters**: the Extreme Treatment Effect `ETE = ξ₁ − ξ₀` at the population level and the Conditional Extreme Treatment Effect `CETE(x) = ξ₁(x) − ξ₀(x)` at the individual level.

The main contributions are:

1. **New estimands** (ETE/CETE) framed via the tail-decay rate, with two motivations (a regular-variation / survival-ratio view and a point-process / expected-exceedances view).
2. **A weaker identifying assumption, "tail unconfoundedness"**,  requiring only the *tail* of the potential outcome (not the full outcome) to be conditionally independent of treatment, proven strictly weaker than standard unconfoundedness (Prop. 3.6), with identifiability (Prop. 3.9) and consistency/asymptotic normality (Remark 3.10).
3. **An estimation pipeline**: an `ε-max-sampler` ("block maxima in space") that forms surrogate block-maxima from neighbourhoods of similar individuals (k-means clusters), plus neural parameterizations of the GEV (reusing S-/T-learner, TARNet, CFR) trained by GEV likelihood; the population ETE is obtained by marginalizing the learned conditionals to remove selection bias.
4. **Experiments** on a synthetic 1-D design and three semi-synthetic IHDP variants (Gumbel/Fréchet/Weibull domains), showing the proposed ETE estimator is approximately unbiased where the naive block-maxima estimator is not.

_____

## Strengths and Weaknesses

### Strengths

- **S1: Clear, well-motivated problem.** The observation that a treatment can be beneficial on average (negative ATE) while increasing the rate of rare catastrophic outcomes (positive ETE) is compelling and clearly illustrated (Fig. 1). It is a genuine gap in the ATE/CATE-centric literature.
- **S2: A sensible estimand and a real theoretical contribution.** Targeting the shape parameter `ξ` rather than an extreme quantile or extreme mean is well-justified: it is the descriptor that remains finite and meaningful precisely in the heavy-tailed regime where moments fail. The **tail unconfoundedness** relaxation is intuitive and is the an interesting result in the paper.
- **S3: Honest handling of the single-observation problem.** The `ε-max-sampler` is a reasonable and practical answer to "you only see each individual once," with supporting theory (Props. C.1–C.2).
- **S4: Selection-bias correction is demonstrated.** The synthetic results (Figs. 2–3) convincingly show the conditional model + marginalization removes the bias of the naive estimator.

### Weaknesses

- **W1: Scope vs. motivation: the framework as stated does not cover many standard count outcomes common in adverse-event settings.** Everything rests on the outcome lying in the max-domain of attraction (MDA) of a *non-degenerate* GEV (Sec. 3.1). The paper motivates the method with serious adverse events, but many such outcomes are recorded as **counts** (e.g., number of events above a threshold), whereas standard light-tailed count laws such as Poisson, geometric, and negative binomial are in **no** classical MDA, so `ξ` (hence ETE/CETE) is not defined for them as stated. This is the classical result that integer-valued laws need the *long-tailed* property `F̄(k+1)/F̄(k)→1` to lie in a classical MDA, which these laws fail (Anderson 1970; Shimura 2012). The paper would be stronger if it stated the continuous/non-lattice scope explicitly, or otherwise clarified which count-valued outcomes are admissible. If the authors wish to extend the framework to count outcomes, a constructive route exists: discrete-EVT replaces the continuous GEV/GPD with a discrete generalized-Pareto / generalized-Zipf peaks-over-threshold model (Hitz, Davis & Samorodnitsky 2024), which has been used for count data.
- **W2: The tail-unconfoundedness illustrations (Appendix D) are weak.** The discrete Bernoulli example shows only that `P(Z_1^n=1 | T=t,X)=1-t^n → 1`, so the sample maximum converges to a **degenerate point mass at 1**, not to a non-degenerate GEV limit. This doesn't  match Sec. 3.1, which assumes non-degenerate GEV limits for the potential outcomes and their conditionals. The Beta example also appears problematic: under the standard Beta parameterization, the upper-tail Weibull index should depend on the second shape parameter (`β=t+1`), unless the authors are using a nonstandard convention.
- **W3: Tail unconfoundedness is genuinely weaker, but not obviously easier to defend in practice.** The authors are right that requiring only the *tail* (not the full outcome) to be conditionally independent of treatment is formally weaker (Prop. 3.6), and this is a real contribution. The caveat I would raise is that it remains untestable and can even be *harder* to reason about: domain experts can sometimes argue "U affects the outcome," but rarely "U affects the *tail index*." In the headline medical setting, the latent drivers of catastrophic tails (genetics, frailty, comorbidities) are often unmeasured, tail-relevant, *and* correlated with treatment. (optional) A sensitivity analysis quantifying how ETE/CETE move under a *bounded* violation would help; the marginal sensitivity model (Tan 2006) with percentile-bootstrap intervals (Zhao, Small & Bhattacharya 2019) is one ready-made template, and a tail-specific version would be a nice addition.
- **W4: Positioning relative to closely related work could be strengthened (not a novelty demand).** EVT-for-causal-inference has become an active area, and a few published works would help readers place the contribution. Most directly, **Deuber, Li, Engelke & Maathuis (2024)** study causal inference in the tails via EVT with extrapolation **beyond the data range** (binary treatment, a "causal Hill" estimator) *with* asymptotic normality, a variance estimator (confidence intervals), and a real-data application; a sentence or two contrasting their extremal-*quantile* target with this paper's *shape-parameter* difference would sharpen the contribution. The recent survey **Chavez-Demoulin & Mhalla (2024)** is a natural umbrella citation. Two further methodological neighbours are worth a mention: **Pasche & Engelke (2024)** (a neural network outputting covariate-dependent EVT tail parameters `σ(x), ξ(x)`; see also W7) and **Sasaki & Wang (2022)** (conditional extreme quantiles from nearest-neighbour tail observations — a close cousin of the `ε-max-sampler`; see W5, W6). TMLR does not require novelty; I raise these only so the contribution is not inadvertently over-stated relative to existing work.
- **W5: The empirical evaluation reads as a solid proof-of-concept rather than a comprehensive study.**
  - *External baselines.* The only ETE comparator is the naive block-maxima estimator. I recognize there is no exact apples-to-apples competitor (no prior method targets the conditional shape-parameter difference with one observation per individual), so the naive estimator is a reasonable reference; still, a quantile-TE method such as Deuber, Li, Engelke & Maathuis (2024), at least in the heavy-tailed setting, would give readers a sense of where the approach sits relative to the field.
  - *Uncertainty on the headline ETE.* Table 2 reports point values with no error bars, even though the paper proves asymptotic normality (Remark 3.10) and Table 1 (CETE) does report seed variability. Confidence intervals or seed spread on Table 2 would directly support the inferential claims.
  - *Bounded-tail case.* IHDP (Weibull) improves only `0.931 → 0.827`, with a still-large absolute error; a sentence on why the bounded case is harder would help readers.
  - *Backbone selection.* Only one backbone, CFR(Wass), is carried into Table 2, despite S-learner performing best on IHDP original.
  - *Simulated ground truth.* All ground truth is simulated; a real-data illustration (even without known truth) would strengthen the "efficacy" claim, or the claim could be softened to "proof of concept."
- **W6: The `ε`/`K` choice hides a bias-variance trade-off that is neither characterized nor guided.** The block radius `ε` (equivalently block size `m`, cluster count `K=n/m`) behaves like a kernel bandwidth: larger blocks reach deeper into the tail but mix heterogeneous individuals (bias `≤ Lε`, Prop. C.2) and yield fewer block-maxima (higher variance); smaller blocks do the reverse. This is the classical block-size trade-off, stated in **Coles (2001, §3.3.1)**, which the paper already cites, and since made operational (e.g. Ferreira & de Haan 2015; Dkengne et al. 2020; the finite-`k` nearest-neighbour version of Sasaki & Wang 2022). The paper supplies the ingredients (the `Lε` bound; a `K`-ablation, Fig. 6) but stops short of a selection rule or rate; a practical default would help. Relatedly, the "neighbours behave like i.i.d. copies" premise weakens as covariate dimension grows, which is worth a comment given IHDP's 25 features (the N/D ablation stops at `D=10`).
- **W7: Sensitivity to the known-`μ,σ` simplification is untested.** Experiments fix `μ, σ` as known and learn only `ξ(x)`, citing instability of full GEV fitting. Fixing location/scale is a common and defensible practical choice, and the shape parameter is what the estimand depends on; still, the sensitivity of ETE/CETE to estimated or misspecified `μ, σ` would be good to report. If the authors would like to learn all three parameters, the orthogonal reparameterization of Pasche & Engelke (2024) (`ν = σ(ξ+1)`, which diagonalizes the Fisher information) is one way to address the instability they cite.

**Additional Comments:**

### References cited in this review (not in the paper)

- Anderson, C. W. (1970). Extreme value theory for a class of discrete distributions with applications to some stochastic processes. *J. Appl. Probab.* 7(1), 99–113.
- Chavez-Demoulin, V., & Mhalla, L. (2024). Causality and Extremes. arXiv:2403.05331.
- Deuber, D., Li, J., Engelke, S., & Maathuis, M. H. (2024). Estimation and Inference of Extremal Quantile Treatment Effects for Heavy-Tailed Distributions. *JASA* 119(547), 2206–2216.
- Dkengne, P. S., Girard, S., & Ahiad, S. (2020). An automatic procedure to select a block size in the continuous GEV model estimation. (hal-02952279.)
- Ferreira, A., & de Haan, L. (2015). On the block maxima method in extreme value theory: PWM estimators. *Ann. Statist.* 43(1), 276–298.
- Gnecco, N., Meinshausen, N., Peters, J., & Engelke, S. (2021). Causal discovery in heavy-tailed models. *Ann. Statist.* 49(3), 1755–1778.
- Hitz, A. S., Davis, R. A., & Samorodnitsky, G. (2024). Discrete Extremes. *J. Data Science* 22(4), 524–536.
- Pasche, O. C., & Engelke, S. (2024). Neural networks for extreme quantile regression with an application to forecasting of flood risk. *Ann. Appl. Stat.* 18(4), 2818–2839.
- Sasaki, Y., & Wang, Y. (2022). Fixed-k Inference for Conditional Extremal Quantiles. *J. Bus. Econ. Stat.* 40(2), 829–837.
- Shimura, T. (2012). Discretization of distributions in the maximum domain of attraction. *Extremes* 15(3), 299–317.
- Tan, Z. (2006). A distributional approach for causal inference using propensity scores. *JASA* 101(476), 1619–1637.
- Zhao, Q., Small, D. S., & Bhattacharya, B. B. (2019). Sensitivity analysis for inverse probability weighting estimators via the percentile bootstrap. *JRSS-B* 81(4), 735–761.

**Audience:**

Yes

**Audience Explanation:**

The combination of EVT and causal inference, the shape-parameter estimand, and the tail-unconfoundedness relaxation will interest researchers working on causal inference, risk/extremes, and treatment-effect estimation. There is clearly "something to be learned" for a segment of TMLR's audience.

**Broader Impact Concerns:**

One caution, because tail unconfoundedness is untestable and (per W3) may fail in exactly the high-stakes medical settings used to motivate the method, the paper could note that ETE/CETE estimates should not drive safety decisions without domain validation of the assumptions.

**Claims And Evidence:**

Yes

**Claims Explanation:**

**Partially yes.**

The theoretical claims appear sound and are supported. Two gaps would benefit from attention before the claims are fully backed: (i) the framework is presented as applicable to the motivating adverse-event settings, but its GEV/MDA requirement excludes the count outcomes common there unless the scope is stated (W1–W2); and (ii) "we demonstrate the efficacy of our approach" is a touch stronger than the current evidence supports, given the absence of an external comparison, error bars on the ETE table, and any real-data validation (W5). Both are fixable via Requested Changes 1–3.

**Requested Changes:**

## Requested Changes

**Critical:**

1. **State the scope explicitly and reconcile it with the motivation (W1).** Add an assumption/remark that outcomes must lie in a non-degenerate GEV domain, note that this **excludes count/lattice outcomes** (Poisson/geometric/neg-binomial; Anderson 1970; Shimura 2012), and either (a) restrict the motivating claims to continuous severity outcomes, or (b) point to the discrete-EVT route (Hitz, Davis & Samorodnitsky 2024) for count data.
2. **Repair or replace the Bernoulli example in Appendix D (W2)** so the tail-unconfoundedness illustration is consistent with the paper's own non-degenerate-GEV assumption; ideally add an example with a *mechanism* for why a confounder affects the body but not the tail.
3. **Tighten the link between claims and evidence (W4, W5):** add a brief discussion of the closest related work (at minimum Deuber et al. 2024, the Chavez-Demoulin & Mhalla 2024 survey, Pasche & Engelke 2024, and Sasaki & Wang 2022); add at least one external comparison *or* report uncertainty (CIs/seed spread) on the ETE results in Table 2; and either add a real-data illustration or soften "demonstrate the empirical efficacy" to "proof of concept."

**Would strengthen (not blocking):**

1. A sensitivity analysis quantifying how ETE/CETE degrade under bounded violations of tail unconfoundedness (W3), e.g. a tail-specific marginal sensitivity model with percentile-bootstrap intervals (Tan 2006; Zhao, Small & Bhattacharya 2019).
2. A bias–variance discussion and a practical rule/rate for choosing `m`/`K` (drawing on Coles 2001 §3.3.1; Ferreira & de Haan 2015; Dkengne et al. 2020; Sasaki & Wang 2022), plus reported sensitivity of results to `K` and to dimension `D` (W6); the N/D ablation currently stops at `D=10` while IHDP has 25 features.
3. Report sensitivity to the known-`μ,σ` simplification, or adopt the orthogonal GEV reparameterization of Pasche & Engelke (2024) to fit all three parameters (W7).
4. Clarify whether tail unconfoundedness on the limit `Z_t` suffices to justify the finite-sample `ε-max-sampler`, or whether a finite-`m` analogue is needed.

---

### Review · Reviewer_zhjV · 2026-07-01

**Summary Of Contributions:**

This paper introduces a new framework based on Extreme Value Theory to estimate how treatments affect the extreme outcomes of a distribution, which is useful for calculating severe risks in areas like healthcare. The authors define new metrics called the Extreme Treatment Effect (ETE) and Conditional Extreme Treatment Effect (CETE) using differences in distribution tail decay rates. They also introduce a weaker "tail unconfoundedness" condition to prove that these effects can be identified even when some background data is missing.

Pros:

- The paper tackles the tail risks to measure extreme, high-risk outcomes where the data is scarce
- The paper defines intuitive parameters that help quantify exactly how much more likely a treatment is to cause extreme events compared to a control group.

Cons:
- Both the method and the experiments assume that the distribution's location and scale parameters are already known. This is an extreme simplification that offloads the severe optimization instabilities found in real-world tail estimation, which significantly reduces the applicability of the method.
- The synthetic datasets were modified to inject the extreme-value noise into the outcomes, it artificially creates the advantage for this specific model.
- The paper failed to compare to more recent extreme treatment models or robust, non-parametric extreme quantile frameworks.

**Audience:**

No

**Audience Explanation:**

The paper's lack of evidence to its claim and the unclear novelty (due to the lack of comparison to the latest competitors) makes it less attractive to the audience.

**Claims And Evidence:**

No

**Claims Explanation:**

The empirical evidence provided in the paper is not convincing or representative of a rigorous real-world validation due to these flaws:
1. The authors claim their framework works well under treatment selection bias. However, they only test it on semi-synthetic datasets where they manually injected the exact GEV-tail noise that their model is built to look for. This gives the model an unfair advantage and does not prove it can handle real-world, unmapped heavy tails.
2. The authors admit that simultaneously fitting all three GEV parameters ($\mu, \sigma, \xi$) is numerically unstable. To bypass this, they assume the location ($\mu$) and scale ($\sigma$) parameters are already perfectly known during testing. Because these parameters are never known in real life, the results represent an idealized upper bound rather than real-world application.
3. The paper claims standard quantile methods are too unstable for extreme regimes. However, it fails to compare against actual extreme quantile treatment effect (QTE) models. For example, non-parametric Tail-Calibrated Inverse Estimating Equation (TIEE) framework [R1].
4. The paper claims to offer a completely "new framework" for this task. However, it fails to cite, discuss, or compare against direct competitors, such as the Estimation of Treatment Effects in Extreme and Unobserved Data framework [R2] from NeurIPS 2025. [R2] already established neural network extrapolation for extreme treatment effects under selection bias using multivariate regular variation.

Refs:
- [R1] Li, Mengran, and Daniela Castro-Camilo. "Tail-Calibrated Estimation of Extreme Quantile Treatment Effects." arXiv preprint arXiv:2603.23309 (2026).
- [R2] Tan, Jiyuan, Vasilis Syrgkanis, and Jose Blanchet. "Estimation of Treatment Effects in Extreme and Unobserved Data." Advances in Neural Information Processing Systems 38 (2026): 113612-113638.

**Requested Changes:**

1. Include the discussion and both methodological and empirical comparison to the missing citations, especially the directly related modern methods.
2. Increase the benchmark dataset variety and the simulation realism to validate the method's applicability in real-world scenarios

---

### Review · Reviewer_tWTq · 2026-07-16

**Summary Of Contributions:**

The paper develops a framework for estimating causal effects of treatment on extreme-value behaviour using GEV distributions, which arise as limiting distributions of suitably normalized maxima of collections of $n$ IID random variables as $n\to \infty$. They define two estimands in terms of the GEV shape parameter (ETE and CETE) and argue that these estimands are informative about the effect of treatment on the asymptotic tail behaviour of the counterfactual outcome distributions. They then formalize a ``tail unconfoundedness'' condition for identifiability, which requires the limiting extreme-value law associated with $p(Y(t)\mid X,T)$ to be invariant to the value of $T$. They show that the usual conditional ignorability criterion implies this condition, and claim that this condition can be strictly weaker. They then formulate a likelihood-based estimator of the conditional GEV law $p(Z(t)\mid X)$ via its shape function $\xi_t(x)$, from which the CETE can be constructed. The average conditional distributions are then averaged over $P(X)$, and samples from the resulting marginal distribution are used to fit a marginal GEV by maximum likelihood (this can be interpreted as an approximate KL projection onto the family of marginal GEV distributions) to yield an estimator of the ETE. They develop a clustering-based approach for turning samples of $Y(t)$ into approximate samples of conditional block maxima, which are then treated as approximately GEV-distributed. They conclude by analyzing the performance of their estimators in several synthetic and semi-synthetic experiments.

**Audience:**

Yes

**Audience Explanation:**

I think the paper's method and findings will be of clear interest and use to the causal inference community. The literature is flooded with methods targeting average effects, while analyzing how treatments affect extreme-value behaviour is clearly understudied and should be of direct interest in applications where one wishes to analyze the extent to which treatment may or may not induce rare adverse consequences.

The proposed approach provides a clean parametric alternative to non-parametric approaches to estimating tail effects due to the survival ratio identity
$$\frac{S_0(y)}{S_1(y)}\in RV\left(\xi_1-\xi_0\right)$$
(or reciprocal version - see above comment).
Since non-parametric estimation of tail-behaviour is known to be challenging, this approach is likely to be of direct interest and use in these applications.

The main limitations to the paper that may reduce interest from the wider causal community is:
1. one does not know exactly how the tail behaviour changes, as one does not recover the individual survival functions themselves using this approach.
2. most of the paper assumes a known location/scale parameter (and the paper notes challenges optimizing these parameters), which is a strong assumption that rarely (if ever) holds in practice.

**Claims And Evidence:**

No

**Claims Explanation:**

The manuscript is clearly well written and the narrative is straightforward to follow (except for one place where a new section break may help the flow of the paper - see requested changes). The main issue with the current manuscript's strength/accuracy of evidence are (i) some mathematical issues that need addressing and (ii) that some relevant baselines may be missing from the empirical evaluation.

**Mathematical Issues**

1. **No ATE in Example:** In the motivating example in Figure 1, using the GEV parameterization in Equation (1), the mean exists only for $\xi<1$. Since the treatment distribution is assigned $\xi_1=1.5$, this would mean the true ATE does not exist. I imagine the authors are reporting the empirical ATE, but this example should be clarified or fixed.

2. **Ill-defined counting process:** In remark 3.4 $N_{Y_t}(R)=\sum_{i=1}^{\infty}\delta_{\{Y_t^{(i)}\in R\}}$ is a raw infinite count, so whenever $P(Y_t\in R)>0$, this would be infinite almost surely and so have infinite expectation whenever the outcomes are iid over $i$.

3. **Conceptual query about $X$:** I am a bit confused by what $X$ represents when coupled with $(Z_t,T)$ in a single probability space. if $Z_t$ represents the limiting distribution of a transformed maximum of $n$ iid potential outcomes $Y_1(t) ,...,Y_n(t)$ (suppose the limit holds a.s. too for convenience), then is there a single $X$ or instead one per unit $(X_i)^{i\geq 1}$? One could set $X:= (X_i)^{i \geq 1}$ and then condition on $X=(x_i)^{i \geq 1}$, where $x_i=x_j=x$ for all sequence elements (which would be consistent with the sampling procedure used in Defn 3.8), but I think this needs to be clarified.

4. **Asymptotic Normality Proof:** The continuous-mapping and delta-method steps are straightforward once consistency and asymptotic normality of $\widehat\xi_0$ and $\widehat\xi_1$ have been established. However, the proof simply assumes these properties. Since the proposed estimator involves cluster-based pseudo-maxima, a conditional neural-network likelihood, and a subsequent generated-sample marginal GEV fit, standard iid GEV-MLE results do not apply off the shelf here. The authors therefore should explicitly establish consistency and asy-normality for their actual estimator.

5. **Compatibility of assumptions in Appendix C**: The assumptions of Propositions C.1 and C.2 seem to be intended to hold jointly (to analyze conditions for convergence of the MDA). However, they seem incompatible except in degenerate cases. To see this, note that at any accumulation point $x$, a.s. Lipschitz continuity implies $Y(x_n)\to Y(x)$ almost surely whenever $x_n\to x$. However, if $Y(x_n)\perp Y(x)$ for every $n$, then for any bounded continuous $f$
$$\mathbb{E}[f(Y(x))^2]=\lim_{n\to\infty}\mathbb{E}[f(Y(x_n))f(Y(x))]=\lim_{n\to\infty}\mathbb{E}[f(Y(x_n))]\mathbb{E}[f(Y(x))]=\mathbb{E}[f(Y(x))]^2,$$
which implies that $Y(x)$ is almost surely constant.

*Additional issues*

1. The paper introduces $Z(t)$ as an MDA, however there is no guarantee that a (a.s.) pointwise limit of the affine transformed maxima sequence exists. For good practice, the paper should make clearer that $Z(t)$ is chosen as an arbitrary random variable representing the limiting GEV distribution.

2. CETE is more "group-specific" (fixed X) than "individual level" (fixed unit i).

3. Under the GEV convention in eq(1), $\xi$ is the extreme-value index, whereas the corresponding Pareto tail exponent is $1/\xi$. Thus, in the Fréchet case wouldnt we have
$S_t(y)=y^{-1/\xi_t}L_t(y)$
rather than $S_t(y)=y^{-\xi_t}L_t(y)$? And so then we would have $\frac{S_0(y)}{S_1(y)}\in RV\left(\frac{1}{\xi_1}-\frac{1}{\xi_0}\right)$ instead of the reciprocal?

4. The Bernoulli tail unconfoundedness example in has a degenerate limiting maximum, so doesn't really demonstrate the strict weakness in the relevant non-degenerate GEV setting. Also, in the Beta example I think the upper-tail index of $\mathrm{Beta}(\alpha,\beta)$ is $\xi=-1/\beta$, not $-1/\alpha$, so for this example to show the unconfoundedness property one would need to set $\alpha = t+1$, rather than $\beta$.

**Lack of Baselines in Empirical Evaluation**

One of the early motivations for the paper's parametric approach is that data in the tails are rare (thus implying that, by contrast, a non-parametric approach to estimate the tails may be harder). I think this is a good motivation, however, the experiments only involve a comparison between the naive (clearly biased) ETE estimator and the proposed projection approach in the paper (using the estimated CETE / conditional GEV model).  In order to assess the strengths of the parametric EVT-based approach used in this paper, the approach should also ideally be compared to a non-parametric approach (e.g. CDF-based) in a setting where both methods can be used to infer the same effect of interest in the data. For example, one could evaluate the accuracy of both methods in predicting whether an appropriate metric of "tail-heaviness" increases or decreases for $Y(1)$ vs $Y(0)$.

**Requested Changes:**

1. **Paper Layout**: After proposition 3.6, I would suggest starting a new section on estimation, since it is currently all under 3.3 (Tail unconfoundedness and identifiability).

2. **Mathematical Results**: Could the authors address the mathematical issues raised above.

3. **ETE Estimator:** Notwithstanding my concern above about $X$ vs $(X_i)_{i \geq1}$, if I understand correctly, the procedure used to recover the ETE can be viewed as solving

$$\widehat\theta_t=\arg\min_\theta\operatorname{KL}(\widehat Q_t,P_\theta),\qquad \widehat Q_t=\int \widehat P(\cdot\mid T=t,X=x)d\widehat P_X(x)$$

where $P_\theta$ is a marginal GEV model and the ETE is the difference between the fitted shape parameters for $t=1$ and $t=0$. This highlights a connection with recent work on debiased estimation of counterfactual distributions [1,2,3,4], where one similarly defines a finite-dimensional projection
$$\theta_t^\star=\arg\min_{\theta}D(Q_t,P_\theta),\qquad Q_t=\int P(\cdot\mid T=t,X=x)dP_X(x),$$
but estimates this projection using debiased or orthogonal procedures rather than a direct plug-in estimator. I think it would greatly strengthen the paper if the authors could make this connection explicit, and (if possible) implement the debiased/one-step estimator for the ETE in the binary treatment case - or otherwise comment on the challenges of doing so; e.g., due to the fact that here the distribution is the MDA $Z(t)$ not $Y(t)$ directly.

4. **Known location and scale parameter:** I would strongly encourage the authors to improve the computational approach / optimization algorithm uses so that the scale and location parameter can be at least semi-reliably estimated.

5. **Empirical Baselines**: Could the authors compare against non-parametric approaches in a task where all methods can be used to answer a common causal query about the effect of treatment on tail behaviour? (e.g., binary yes/no for tail-heaviness change).